# Optical flow models as an open benchmark for radar-based precipitation nowcasting (rainymotion v0.1)

Georgy Ayzel[1], Maik Heistermann[1], and Tanja Winterrath[2]

[1]Institute for Environmental Sciences and Geography, University of Potsdam, Potsdam, Germany
[2]Deutscher Wetterdienst, Department of Hydrometeorology, Offenbach, Germany

**Correspondence:** Georgy Ayzel (ayzel@uni-potsdam.de)

**Abstract.** Quantitative precipitation nowcasting (QPN) has become an essential technique in various application contexts, such as early warning or urban sewage control. A common heuristic prediction approach is to track the motion of precipitation features from a sequence of weather radar images, and then to displace the precipitation field to the imminent future (minutes to hours) based on that motion, assuming that the intensity of the features remains constant ("Lagrangian persistence"). In that context, "optical flow" has become one of the most popular tracking techniques. Yet, the present landscape of computational QPN models still struggles with producing open software implementations. Focusing on this gap, we have developed and extensively benchmarked a stack of models based on different optical flow algorithms for the tracking step, and a set of parsimonious extrapolation procedures based on image warping and advection. We demonstrate that these models provide skillful predictions comparable with or even superior to state-of-the-art operational software. Our software library ("rainymotion") for precipitation nowcasting is written in the Python programming language, and openly available at GitHub (https://github.com/hydrogo/rainymotion). That way, the library may serve as a tool for providing fast, free and transparent solutions that could serve as a benchmark for further model development and hypothesis testing – a benchmark that is far more advanced than the conventional benchmark of Eulerian persistence commonly used in QPN verification experiments.

## 1 Introduction

How much will it rain within the next hour? The term "quantitative precipitation nowcasting" refers to forecasts at high spatio-temporal resolution (60-600 seconds, 100-1000 meters) and short lead times of only a few hours. Nowcasts have become important for broad levels of the population for planning various kinds of activities. Yet, they are particularly relevant in the context of early warning of heavy convective rainfall events, and their corresponding impacts such as flash floods, landslides, or sewage overflow in urban areas.

While recent advances in numerical weather prediction (NWP) allow us to forecast atmospheric dynamics at very high resolution (Bauer et al., 2015), computational costs are typically prohibitive for the requirements of operational nowcasting applications with frequent update cycles. Furthermore, the heuristic extrapolation of rain field motion and development, as observed by weather radar, still appears to outperform NWP forecasts at very short lead times (Berenguer et al., 2012; Jensen et al., 2015; Lin et al., 2005). Today, many precipitation nowcasting systems are operational at regional or national scales,

utilizing various radar products, algorithms, and blending techniques in order to provide forecasts up to 1-3 hours, for example: ANC (Mueller et al., 2003), MAPLE (Germann and Zawadzki, 2002), RADVOR (Winterrath et al., 2012), STEPS (Bowler et al., 2006), STEPS-BE (Foresti et al., 2016), and SWIRLS (Cheung and Yeung, 2012; Woo and Wong, 2017). For an extensive review of existing operational systems, please refer to Reyniers (2008).

A variety of radar-based precipitation nowcasting techniques can be classified into three major groups based on assumptions we make regarding precipitation field characteristics (Germann and Zawadzki, 2002). The first group – climatological persistence – provides nowcasts by using climatological values (mean or median). The second group – Eulerian persistence – is based on using the latest available observation as a prediction, and is thus independent from the forecast lead time. The third group – Lagrangian persistence – allows the extrapolation of the most recent observed precipitation field under the assumption that
intensity of precipitation features and the motion field are persistent (Germann and Zawadzki, 2002; Woo and Wong, 2017). In addition, we can classify nowcasting methods based on how predictive uncertainty is accounted for: In contrast to deterministic approaches, ensemble nowcasts attempt to account for predictive uncertainty by including different realizations of the motion field and the evolution of rainfall intensity itself (Berenguer et al., 2011). In this study, we focus our model development around the group of Lagrangian persistence models which provide deterministic precipitation nowcasts.

Lagrangian methods consist of two computational steps: tracking and forecasting (extrapolation) (Austin and Bellon, 1974). In the tracking step, we compute a velocity field from a series of consecutive radar images, either on a per pixel basis (Germann and Zawadzki, 2002; Grecu and Krajewski, 2000; Liu et al., 2015; Zahraei et al., 2012), or for contiguous objects (Zahraei et al., 2013). In the second step, we use that velocity field to advect the most recent rain field, i.e. to displace it to the imminent future based on its observed motion. That step has been implemented based on semi-Lagrangian schemes (Germann and Zawadzki, 2002), interpolation procedures (Liu et al., 2015), or mesh-based models (Bellerby, 2006; Zahraei et al., 2012). Different
algorithms can be used for each step, tracking and forecasting, in order to compute an ensemble forecast (Berenguer et al., 2011; Foresti et al., 2016; Grecu and Krajewski, 2000).

One of the most prominent techniques for the tracking step is referred to as "optical flow". The original term was inspired by the idea of an *apparent* motion of brightness patterns observed when a camera or the eyeball is moving relative to the objects
(Horn and Schunck, 1981). Today, optical flow is often understood as a group of techniques to infer motion patterns or velocity fields from consecutive image frames, e.g. in the field of precipitation nowcasting (Bowler et al., 2004; Liu et al., 2015; Woo and Wong, 2017). For the velocity field estimation, we need to accept both the brightness constancy assumption and one of a set of additional optical flow constraints (OFC). The spatial attribution of OFC marks the two main categories of optical flow models: local (differential) and global (variational) (Cheung and Yeung, 2012; Liu et al., 2015). Local models try to set an OFC
only in some neighborhood, while global models apply an OFC for a whole image. There is also a distinct group of spectral methods where the Fourier transform is applied to the inputs, and an OFC is resolved in the spectral (Fourier) domain (Ruzanski et al., 2011). Bowler et al. (2004) introduced the first local optical flow algorithm for precipitation nowcasting, and gave rise to a new direction of models. Bowler's algorithm is the basis of the STEPS (Bowler et al., 2006) and STEPS-BE (Foresti et al., 2016) operational nowcasting systems. Liu et al. (2015) proposed using a local Lucas–Kanade optical flow method (Lucas and
Kanade, 1981) independently for each pixel of satellite imagery because they found it outperformed a global Horn–Schunck

(Horn and Schunck, 1981) optical flow algorithm in the context of precipitation nowcasting from infrared satellite images. Yeung et al. (2009), Cheung and Yeung (2012), and Woo and Wong (2017) used different global optical flow algorithms (Bruhn et al., 2005a; Wong et al., 2009) for establishing the SWIRLS product for operational nowcasting in Hong-Kong.

Hence, for around two decades, optical flow algorithms have been doing their best for state-of-the-art operational nowcasting systems around the globe. Should research still care about them? It should. . . and the reason is that – despite the abundance of publications about different flavours of optical flow techniques for nowcasting applications – an open and transparent benchmark model is yet not available, except for the most trivial one: Eulerian persistence.

That is all the more surprising since open source implementations of fundamental optical flow algorithms (Brox et al., 2004; Bruhn et al., 2005b) have been around for up to 20 years – with the OpenCV library (https://opencv.org) just being the most widely known. Such libraries provide efficient implementations of various optical flow algorithms for a vast number of research and application contexts. Yet, none can be applied in the QPN context out of the box – without the need to address additional and specific challenges such as underlying assumptions and constraints of velocity fields, pre- and postprocessing steps, or model parameterization and verification.

The aim of this paper is thus to establish a set of benchmark procedures for quantitative precipitation nowcasting as an alternative to the trivial case of Eulerian persistence. This study does not aim to improve the standard of precipitation nowcasting beyond the state-of-the-art, but to provide an open, transparent, reproducible and easy-to-use approach that can compete with the state-of-the-art, and against which future advances can be measured. To that end, we developed a group of models that are based on two optical flow formulations of algorithms for the tracking step – sparse (Lucas and Kanade, 1981) and dense (Kroeger et al., 2016) – together with two parsimonious extrapolation techniques based on image warping and spatial interpolation. These models are verified against Eulerian persistence, as a trivial benchmark, and against the operational nowcasting system of the Deutscher Wetterdienst (the German Weather Service, DWD), as a representative of state-of-the-art models. The different optical flow implementations are published as an open source Python library (*rainymotion*, https://github.com/hydrogo/rainymotion) that entirely relies on free and open source dependencies, including detailed documentation and example workflows (https://rainymotion.readthedocs.io).

The paper is organized as follows. In Section 2, we describe the algorithmic and technical aspects of the suggested optical flow models. Section 3 describes the data we used, and provides a short synopsis of events we used for the benchmark experiment. We report the results in Section 4, and discuss them in various contexts in Section 5. Section 6 provides summary and conclusions.

## 2 Description of the models and the library

The benchmark models developed in this study consist of different combinations of algorithms for the two major steps of Lagrangian nowcasting frameworks, namely tracking and extrapolation (Austin and Bellon, 1974). Table 1 provides an overview of the models. The values of model parameters adopted in the benchmark experiment have been heuristically determined and not yet been subject to systematic optimization. However, the *rainymotion* library provides an opportunity to investigate how

different optical flow model parameters can affect nowcasting results, or how they can be tuned to represent, e.g., the typical range of advection speeds of real precipitation fields. For a description of parameters, please refer to Section S1 in the Supplementary Information or the *rainymotion* library documentation (https://rainymotion.readthedocs.io/).

## 2.1 The Sparse group

The central idea around this group of methods is to identify distinct features in a radar image that are suitable for tracking. In this context, a "feature" is defined as a distinct point ("corner") with a sharp gradient of rainfall intensity. That approach is less arbitrary and scale dependent and thus more universal than classical approaches that track storm cells as contiguous objects (e.g., Wilson et al., 1998) because it eliminates the need to specify arbitrary and scale dependent characteristics of "precipitation features" while the identification of "corners" depends only on the gradient sharpness in a cell's neighborhood. Inside this group, we developed two models that slightly differ with regard to both tracking and extrapolation.

The first model (SparseSD, for Sparse Single Delta) uses only the two most recent radar images for identifying, tracking, and extrapolating features. Assuming that *t* denotes both the nowcast issue time and the time of the most recent radar image, the implementation can be summarized as follows:

1. Identify features in a radar image at time *t-1* using the Shi–Tomasi corner detector (Shi and Tomasi, 1994). This detector determines the most prominent corners in the image based on the calculation of the corner quality measure ($min(\lambda_1, \lambda_2)$, where $\lambda_1$ and $\lambda_2$ are corresponding Eigenvalues of the covariance matrix of derivatives over the neighborhood of 3×3 pixels) at each image pixel (see Section S1 of the Supplementary Information for a detailed description of algorithm parameters);

2. Track these features at time *t* using the local Lucas–Kanade optical flow algorithm (Lucas and Kanade, 1981). This algorithm tries to identify the location of a feature we previously identified at time *t-1* in the radar image at time *t*, based on solving a set of optical flow equations in the local feature neighborhood using the least-squares approach (see Section S1 of the Supplementary Information for a detailed description of algorithm parameters);

3. Linearly extrapolate the features' motion in order to predict the features' locations at each lead time *n*;

4. Calculate the affine transformation matrix for each lead time *n* based on the locations of all identified features at time *t* and *t+n* using the least-squares approach (Schneider and Eberly, 2003). This matrix uniquely identifies the required transformation of the last observed radar image at time *t* so that the nowcast images at times *t+1...t+n* provide the smallest possible difference between the locations of detected features at time *t* and the extrapolated features at times *t+1...t+n*;

5. Extrapolate the radar image at time *t* by warping: for each lead time, the warping procedure uniquely transforms each pixel location of the radar image at time *t* to its future location in the nowcast radar images at times *t+1...t+n*, using the affine transformation matrix. Remaining discontinuities in the predicted image are linearly interpolated in order to obtain nowcast intensities on a grid that corresponds to the radar image at time *t* (Wolberg, 1990).

To our knowledge, this study is the first to apply image warping directly as a simple and fast algorithm to represent advective motion of a precipitation field. In Section S2 of the Supplementary Information, you can find a simple synthetic example which shows the potential of the warping technique to replace an explicit advection formulation for temporal extrapolation.

For a visual representation of the SparseSD model, please refer to Fig. 1.

The second model (Sparse) uses the 24 most recent radar images, and we consider only features that are persistent over the whole period (of 24 time steps). The implementation can be summarized as follows:

1. Identify features on a radar image at time *t-23* using the Shi–Tomasi corner detector (Shi and Tomasi, 1994);

2. Track these features in the radar images from *t-22* to *t* using the local Lucas–Kanade optical flow algorithm (Lucas and Kanade, 1981);

3. Build linear regression models which independently parameterize changes in coordinates through time (from *t-23* to *t*) for every successfully tracked feature;

4. Continue with steps 3-5 of SparseSD.

For a visual representation of the Sparse model, please refer to Fig. 2.

## 2.2 The Dense group

The Dense group of models uses, by default, the Dense Inverse Search algorithm (DIS) – a global optical flow algorithm proposed by Kroeger et al. (2016) – which allows us to explicitly estimate the velocity of each image pixel based on an analysis of two consecutive radar images. The DIS algorithm was selected as the default optical flow method for motion field retrieval because it showed, in our benchmark experiments, a higher accuracy and also a higher computational efficiency in comparison with other global optical flow algorithms such as DeepFlow (Weinzaepfel et al., 2013), and PCAFlow (Wulff and Black, 2015).

We also tested the local Farnebäck algorithm (Farnebäck, 2003), which we modified by replacing zero velocities by interpolation, and by smoothing the obtained velocity field based on a variational refinement procedure (Brox et al., 2004) (please refer to Section S5 in the Supplementary Information for verification results of the corresponding benchmark experiment with various dense optical flow models). However, the *rainymotion* library provides the option to choose any of the specified above optical flow methods for precipitation nowcasting.

The two models in this group differ only with regard to the extrapolation (or advection) step. The first model (Dense) uses a constant-vector advection scheme (Bowler et al., 2004), while the second model (DenseRotation) uses a semi-Lagrangian advection scheme (Germann and Zawadzki, 2002). The main difference between both approaches is that a constant-vector scheme does not allow for the representation of rotational motion (Bowler et al., 2004); a semi-Lagrangian scheme allows for the representation of large-scale rotational movement while assuming the motion field itself to be persistent (Fig. 3).

There are two possible options of how both advection schemes may be implemented: forward in time (and downstream in space) or backward in time (and upstream in space) (Fig. 3). It is yet unclear which scheme can be considered as the most appropriate and universal solution for radar-based precipitation nowcasting, regarding the conservation of mass on the one

hand and the attributed loss of power at small scales on the other hand (e.g., see discussion in Bowler et al., 2004; Germann and Zawadzki, 2002). Thus, we conducted a benchmark experiment with any possible combination of forward vs. backward and constant-vector vs. semi-Lagrangian advection. Based on the results (see Section S6 in the Supplementary Information), we use the backward scheme as the default option for both the Dense and DenseRotation models. However, the *rainymotion* library still provides the option to use the forward scheme, too.

Both the Dense and DenseRotation models utilize a linear interpolation procedure in order to interpolate advected rainfall intensities at their predicted locations to the native radar grid. The interpolation procedure "distributes" the value of a rain pixel to its neighborhood, as proposed in different modifications by Bowler et al. (2004), Liu et al. (2015), and Zahraei et al. (2012). The Dense group models' implementation can be summarized as follows:

1. Calculate a velocity field using the global DIS optical flow algorithm (Kroeger et al., 2016), based on the radar images at time *t-1* and *t*;

2. Use a backward constant-vector (Bowler et al., 2004) or a backward semi-Lagrangian scheme (Germann and Zawadzki, 2002) to extrapolate (advect) each pixel according to the displacement (velocity) field, in one single step for each lead time *t+n*. For the semi-Lagrangian scheme, we update the velocity of the displaced pixels at each prediction time step *n* by linear interpolation of the velocity field to a pixel's location at that time step;

3. As a result of the advection step, we basically obtain an irregular point cloud that consists of the original radar pixels displaced from their original location. We use the intensity of each displaced pixel at its predicted location at time *t+n* in order to interpolate the intensity at each grid point of the original (native) radar grid (Liu et al., 2015; Zahraei et al., 2012), using the inverse distance weighting interpolation technique. It is important to note that we minimize numerical diffusion by first advecting each pixel over the target lead time before applying the interpolation procedure (as in the "interpolate once" approach proposed by Germann and Zawadzki (2002)). That way, we avoid rainfall features to be smoothed in space by the effects of interpolation.

## 2.3 Persistence

The (trivial) benchmark model of Eulerian persistence assumes that for any lead time *n*, the precipitation field is the same as for time *t*. Despite its simplicity, it is quite a powerful model for very short lead times, and, at the same time, its verification performance is a good measure of temporal decorrelation for different events.

## 2.4 The *rainymotion* Python library

We have developed the *rainymotion* Python library to implement the above models. Since the *rainymotion* uses standard format of *numpy* arrays for data manipulation, there is no restriction in using different data formats which can be read, transformed, and converted to *numpy* arrays using any tool from the set of available open software libraries for radar data manipulation (the list is available on https://openradarscience.org). The source code is available in a Github repository (https:

//github.com/hydrogo/rainymotion), and has a documentation page (https://rainymotion.readthedocs.io) which includes installation instructions, model description, and usage examples. The library code and accompanying documentation are freely distributed under the MIT software license which allows unrestricted use. The library is written in the Python 3 programming language (https://python.org) and its core is entirely based on open source software libraries (Fig. 4): $\omega radlib$ (Heistermann et al., 2013), *OpenCV* (Bradski and Kaehler, 2008), *SciPy* (Jones et al., 2018), *NumPy* (Oliphant, 2006), *Scikit-learn* (Pedregosa et al., 2011), and *Scikit-image* (Van der Walt et al., 2014). For generating figures we use the *Matplotlib* library (Hunter, 2007), and we use the *Jupyter notebook* (https://jupyter.org) interactive development environment for code and documentation development and distribution. For managing the dependencies without any conflicts, we recommend to use the *Anaconda* Python distribution (https://anaconda.com) and follow *rainymotion* installation instructions (https://rainymotion.readthedocs.io).

## 2.5 Operational baseline (RADVOR)

The DWD operationally runs a stack of models for radar-based nowcasting and provides precipitation forecasts for a lead time up to 2 hours. The operational QPN is based on the RADVOR module (Bartels et al., 2005; Rudolf et al., 2012). The tracking algorithm estimates the motion field from the latest sequential clutter-filtered radar images using a pattern recognition technique on different spatial resolutions (Winterrath and Rosenow, 2007; Winterrath et al., 2012). The focus of the tracking algorithm is on the meso-$\beta$ scale (spatial extent: 25–250 km) to cover mainly large-scale precipitation patterns, but the meso-$\gamma$ scale (spatial extension: 2.5–25 km) is also incorporated to allow the detection of smaller-scale convective structures. The resulting displacement field is interpolated to a regular grid, and a weighted averaging with previously derived displacement fields is implemented to guarantee a smooth displacement over time. The extrapolation of the most recent radar image according to the obtained velocity field is performed using a semi-Lagrangian approach. The described operational model is updated every 5 minutes and produces precipitation nowcasts at a temporal resolution of 5 minutes and a lead time of 2 hours (RV product). In this study we used the RV product data as an operational baseline and did not re-implement the underlying algorithm itself.

## 3 Verification experiments

### 3.1 Radar data and verification events

We use the so-called RY product of the DWD as input to our nowcasting models. The RY product represents a quality-controlled rainfall depth product that is a composite of the 17 operational Doppler radars maintained by the DWD. It has a spatial extent of 900×900 km and covers the whole area of Germany. Spatial and temporal resolution of the RY product is 1×1 km and 5 minutes, respectively. This composite product includes various procedures for correction and quality control (e.g. clutter removal). We used the $\omega radlib$ (Heistermann et al., 2013) software library for reading the DWD radar data.

For the analysis, we have selected 11 events during the summer periods of 2016 and 2017. These events are selected for covering a range of event characteristics with different rainfall intensity, spatial coverage, and duration. Table 2 shows the studied events. You can also find links to animations of event intensity dynamics in Section S3 of the Supplementary Information.

### 3.2 Verification metrics

For the verification we use two general categories of scores: continuous (based on the differences between nowcast and observed rainfall intensities) and categorical (based on standard contingency tables for calculating matches between boolean values which reflect the exceedance of specific rainfall intensity thresholds). We use the mean absolute error (MAE) as a continuous score:

$$MAE = \frac{\sum_{i=1}^{n} |now_i - obs_i|}{n} \tag{1}$$

where $now_i$ and $obs_i$ are nowcast and observed rainfall rate in the *i*-th pixel of the corresponding radar image, and *n* the number of pixels. To compute the MAE, no pixels were excluded based on thresholds of nowcast or observed rainfall rate.

And we use the critical success index (CSI) as a categorical score:

$$CSI = \frac{hits}{hits + false\ alarms + misses} \tag{2}$$

where *hits*, *false alarms*, and *misses* are defined by the contingency table and the corresponding threshold value (for details see Section S4 of the Supplementary Information).

Following studies of Bowler et al. (2006) and Foresti et al. (2016) we have applied threshold rain rates of 0.125, 0.25, 0.5, 1 and 5 mm h$^{-1}$ for calculating the CSI.

These two metrics inform us about the models' performance from the two perspectives: MAE captures errors in rainfall rate prediction (the less the better), and CSI captures model accuracy (the fraction of the forecast event that was correctly predicted; does not distinguish the source of errors; the higher the better). You can find results represented in terms of additional categorical scores (false alarm rate, probability of detection, equitable threat score) in Section S4 of the Supplementary Information.

### 4 Results

For each event, all models (Sparse, SparseSD, Dense, DenseRotation, Persistence) were used to compute nowcasts with lead times from 5 to 60 minutes (in 5 minute steps). Operational nowcasts generated by the RADVOR system were provided by the DWD with the same temporal settings. An example of nowcasts for lead times 0, 5, 30, and 60 minutes is shown in Fig. 5.

To investigate the effects of numerical diffusion, we calculated, for the same example, the power spectral density (PSD) of the nowcasts and the corresponding observations (bottom panel in Figure 5) using Welch's method (Welch, 1967). Germann and Zawadzki (2002) showed that the most significant loss of power (lower PSD values) occurs at scales between 8 to 64 km. They did not analyze scales below 8 ($2^3$) km because their original grid resolution was 4 km. We extended the spectral analysis to consider scales as small as $2^1$ km. Other than Germann and Zawadzki (2002), we could not observe any substantial loss of power between 8 and 64 km, yet Figure 5 shows that both Dense and Sparse models consistently start to lose power at scales below 4 km. That loss does not depend much on the nowcast lead time, yet, the Sparse group of models loses more power at a lead time of 5 minutes as compared to the Dense group. Still, these results rather confirm Germann and Zawadzki (2002): they

show, as would be expected, that any loss of spectral power is most pronounced at the smallest scales, and disappears at scales about 2-3 orders above the native grid resolution. For the investigated combination of data and models, that implies that our nowcasts will not be able to adequately represent rainfall features smaller than 4 km at lead times of up to 1 hour.

Figure 6 shows the model performance (in terms of MAE) as a function of lead time. For each event, the Dense group of models is superior to the other ones. The RV product achieves an efficiency that is comparable to the Dense group. The SparseSD model outperforms the Sparse model for short lead times (up to 10-15 minutes), and vice versa for longer lead times. For some events (1-4, 6, 10, 11), the performance of the RV product appears to be particularly low in the first 10 minutes, compared to the other models. These events are characterized by particularly fast rainfall field movement.

Figure 7 has the same structure as Fig. 6, but shows the CSI with a threshold value of 1 mm h$^{-1}$. For two events (7 and 10) the RV product achieves a comparable efficiency with the Dense group for lead times beyond 30 minutes. For the remaining events, the Dense group tends to outperform all other methods and the RV product achieves an average rank between models of the Sparse and Dense groups. For the Dense group of models, it appears that accounting for field rotation does not affect the results of the benchmark experiment much – the Dense and DenseRotation models perform very similarly, at least for the selected events and the analyzed lead times. The behavior of the Sparse group models is mostly consistent with the MAE.

Figure 8 shows the model performance using the CSI with a threshold value of 5 mm h$^{-1}$. For the majority of events, the resulting ranking of models is the same as for the CSI with a threshold of 1 mm h$^{-1}$. For events #2 and #3, the performance of the RV product relative to the Dense models is a little bit better, while for other events (e.g. #7), the Dense models outperform the RV product more clearly than for the CSI of 1 mm h$^{-1}$.

Table 3 summarizes the results of the Dense group models in comparison to the RV product for different verification metrics averaged over all the selected events and two lead time periods: 5–30, and 35–60 minutes. Results show that the Dense group always slightly outperforms the RV model in terms of CSI metric for both lead time periods and all analyzed rainfall intensity threshold used for CSI calculation. In terms of MAE, differences between model performances are less pronounced. For the CSI metric, the absolute differences between all models tend to be consistent with increasing rainfall thresholds.

You can find more figures illustrating the models' efficiency for different thresholds and lead times in Section S4 of the Supplementary Information.

## 5 Discussion

### 5.1 Model comparison

All tested models show significant skill over the trivial Eulerian persistence over a lead time of at least one hour. Yet, a substantial loss of skill over lead time is present for all analyzed events, as expected. We have not disentangled the causes of that loss, but predictive uncertainty will always result from errors in both the representation of field motion and the total lack of representing precipitation formation, dynamics, and dissipation in a framework of Lagrangian persistence. Many studies specify a lead time of 30 minutes as a predictability limit for convective structures with fast dynamics of rainfall evolution

(Foresti et al., 2016; Grecu and Krajewski, 2000; Thorndahl et al., 2017; Wilson et al., 1998; Zahraei et al., 2012). Our study confirms these findings.

For the majority of analyzed events, there is a clear pattern that the Dense group of optical flow models outperforms the operational RV nowcast product. For the analyzed events and lead times, the differences between the Dense and the DenseRotation models (or, in other words, between constant-vector and semi-Lagrangian schemes), are negligible. The absolute difference in performance between the Dense group models and the RV product appears to be independent from rainfall intensity threshold and lead time (Table 3), which implies that the relative advance of the Dense group models over the RV product increases both with lead time and rainfall intensity threshold. A gain in performance for longer lead times by taking into account more time steps from the past can be observed when comparing the SparseSD model (looks back five minutes in time) against the Sparse model (looks back two hours in time).

Despite their skill over Eulerian persistence, the Sparse group models are significantly outperformed by the Dense group models for all the analyzed events and lead times. The reason for this behaviour remains yet unclear. It could, in general, be a combination of errors introduced in corner-tracking and extrapolation as well as image warping as a surrogate for formal advection. While the systematic identification of error sources will be subject to future studies, we suspect that the the local features ("corners") identified by the Shi–Tomasi corner detector might not be representative for the overall motion of the precipitation field: the detection focuses on features with high intensities and gradients, the motion of which might not represent the dominant meso-$\gamma$ scale motion patterns.

There are a couple of possible directions for enhancing the performance for longer lead times using the Dense group of models. A first is to use a weighted average of velocity fields derived from radar images three (or more) steps back in time (as done in RADVOR to compute the RV product). A second option is to calculate separate velocity fields for low and high intensity subregions of the rain field, and advect these subregions separately (like proposed in Golding (1998)), or find an optimal weighting procedure. A third approach could be to optimize the use of various optical flow constraints in order to improve the performance for longer lead times, as proposed in Germann and Zawadzki (2002), Bowler et al. (2004), or Mecklenburg et al. (2000). The flexibility of the *rainymotion* software library allows users to incorporate such algorithms for benchmarking any hypothesis, and e.g. implement different models or parameterizations for different lead times. Bowler et al. (2004) also showed a significant performance increase for longer lead times by using NWP model winds for the advection step. However, Winterrath and Rosenow (2007) did not obtain any improvement compared to RADVOR for longer lead times by incorporating NWP model winds into the nowcasting procedure.

## 5.2 Advection schemes properties and effectiveness

Within the Dense group of models, we could not find any significant difference between the performance and PSD of the constant-vector (Dense model) and the Semi-Lagrangian scheme (DenseRotation). That confirms findings presented by Germann and Zawadzki (2002) who found that the constant vector and the modified semi-Lagrangian schemes have very similar power spectra, presumably since they share the same interpolation procedure. The theoretical superiority of the Semi-

Lagrangian scheme might, however, materialize for other events with substantial, though persistent rotational motion. A more comprehensive analysis should thus be subject to future studies.

Interpolation is included in both the post-processing of image warping (Sparse models) and in the computation of gridded nowcasts as part of the Dense models. In general, such interpolation steps can lead to numerical diffusion and thus to the degradation or loss of small-scale features (Germann and Zawadzki, 2002). Yet, we were mostly able to contain such adverse effects for both the Sparse and the Dense group of models by carrying out only one interpolation step for any forecast at a specific lead time. We showed that numerical diffusion was negligible for lead times up to one hour for any model, however, as had been shown in Germann and Zawadzki (2002), for longer lead times these effects can be significant, depending on the implemented extrapolation technique.

## 5.3 Computational performance

Computational performance might be an important criterion for end users aiming at frequent update cycles. We ran our nowcasting models on a standard office PC with an Intel® Core™ i7-2600 CPU (8 cores, 3.4 GHz), and on a standard laptop with an Intel® Core™ i5-7300HQ CPU (4 cores, 2.5 GHz). The average time for generating one nowcast for one hour lead time (at 5 minute resolution) for the Sparse group is 1.5–3 s, and for the Dense group is 6–12 s. The Dense group is computationally more expensive due to interpolation operations implemented for large grids (900×900 pixels). There is also potential for increasing the computational performance of the interpolation.

## 6 Summary and conclusions

Optical flow is a technique for deriving a velocity field from consecutive images. It is widely used in image analysis, and became increasingly popular in meteorological applications over the past 20 years. In our study, we examined the performance of optical flow based models for radar-based precipitation nowcasting, as implemented in the open-source *rainymotion* library, for a wide range of rainfall events using radar data provided by the DWD.

Our benchmark experiments, including an operational baseline model (the RV product provided by the DWD), show a firm basis for using optical flow in radar-based precipitation nowcasting studies. For the majority of the analyzed events, models from the Dense group outperform the operational baseline. The Sparse group of models showed significant skill, yet they performed generally poorer than both the Dense group and the RV product. We should, however, not prematurely discard the group of Sparse models before we have not gained a better understanding of error sources with regard to the tracking, extrapolation and warping steps. It might also be considered to combine the warping procedure for the extrapolation step with the Dense optical flow procedure for the tracking step (i.e. to advect "corners" based on a "Dense" velocity field obtained by implementing one of the dense optical flow techniques). This opens the way for merging two different model development branches in the future releases of the *rainymotion* library.

There is a clear and rapid model performance loss over lead time for events with high rainfall intensities. This issue continues to be unresolved by standard nowcasting approaches, but some improvement in this field may be achieved with using strategies

such as merging with NWP results and stochastic modelling of rainfall field evolution. Admittedly, deterministic nowcasts in a Lagrangian framework do neither account for precipitation intensity dynamics nor for the uncertainties in representing precipitation field motion. At least for the latter, the *rainymotion* library provides ample opportunities to experiment with forecast ensembles, based on various tracking and extrapolation techniques. Furthermore, we suppose that using new data-driven models based on machine and deep learning may increase the performance by utilizing and structuring common patterns in the massive archives of radar data.

We do not claim that the developed models will compete with well-established and excessively tuned operational models for radar-based precipitation nowcasting. Yet, we hope our models may serve as an essential tool for providing a fast, free and open source solution that can serve as a benchmark for further model development and hypothesis testing – a benchmark that is far more advanced than the conventional benchmark of Eulerian persistence.

Recent studies show that open source community-driven software advances the field of weather radar science (Heistermann et al., 2015a, b). Just a few months ago, the *pySTEPS* (https://pysteps.github.io) initiative was introduced "to develop and maintain an easy to use, modular, free and open source python framework for short-term ensemble prediction systems." As another evidence of the dynamic evolution of QPN research over the recent years, these developments could pave the way for future synergies between the *pySTEPS* and *rainymotion* projects – towards the availability of open, reproducible, and skillful methods in quantitative precipitation nowcasting.

*Code and data availability.* The *rainymotion* library is free and open source. It is distributed under the MIT software license which allows unrestricted use. The source code is provided through a GitHub repository https://github.com/hydrogo/rainymotion, the snapshot of the *rainymotion* v0.1 is also available on Zenodo: https://doi.org/10.5281/zenodo.2561583, and the documentation is available on a website https://rainymotion.readthedocs.io. The DWD provided the sample data of the RY product, and it is distributed with the *rainymotion* repository to provide a real case and reproducible example of precipitation nowcasting.

*Author contributions.* GA performed the benchmark experiments, analyzed the data and wrote the manuscript. MH coordinated and supervised the work, analyzed the data and wrote the manuscript. TW assisted in the data retrieval and analysis, and shared her expertise in DWD radar products.

*Competing interests.* The authors declare that they have no conflict of interest.

*Acknowledgements.* Georgy Ayzel was financially supported by Geo.X, the Research Network for Geosciences in Berlin and Potsdam. The authors thank Loris Foresti, Seppo Pulkkinen, and Remko Uijlenhoet for their constructive comments and suggestions that helped to improve

the paper. We acknowledge the support of Deutsche Forschungsgemeinschaft (German Research Foundation) and Open Access Publication Fund of Potsdam University.

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

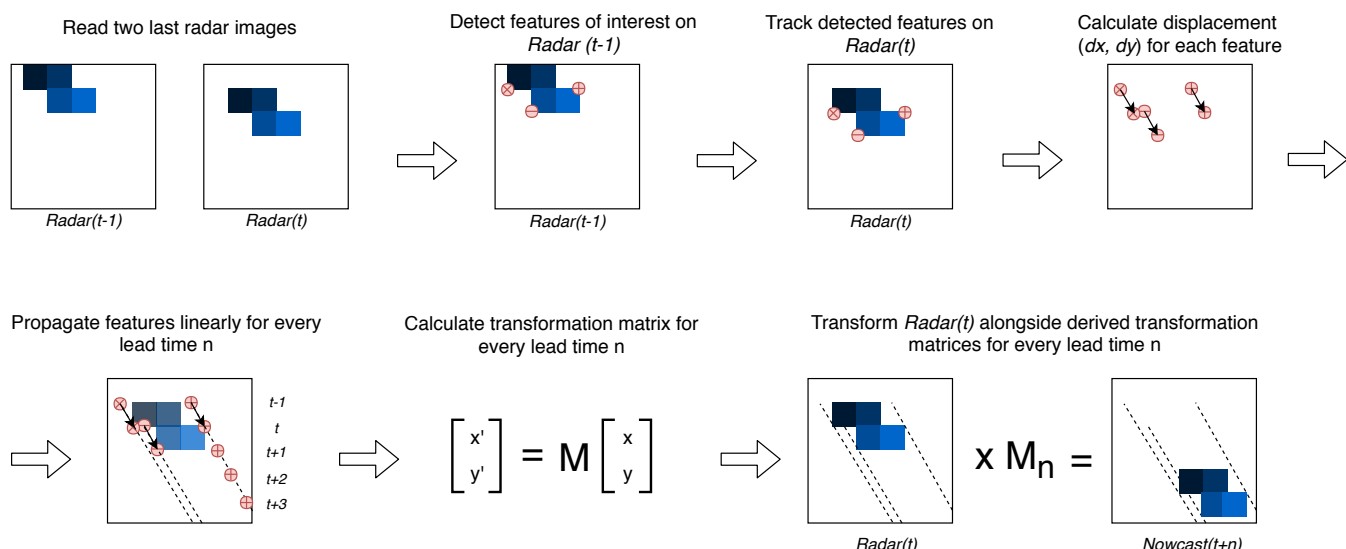

**Figure 1.** Scheme of the SparseSD model

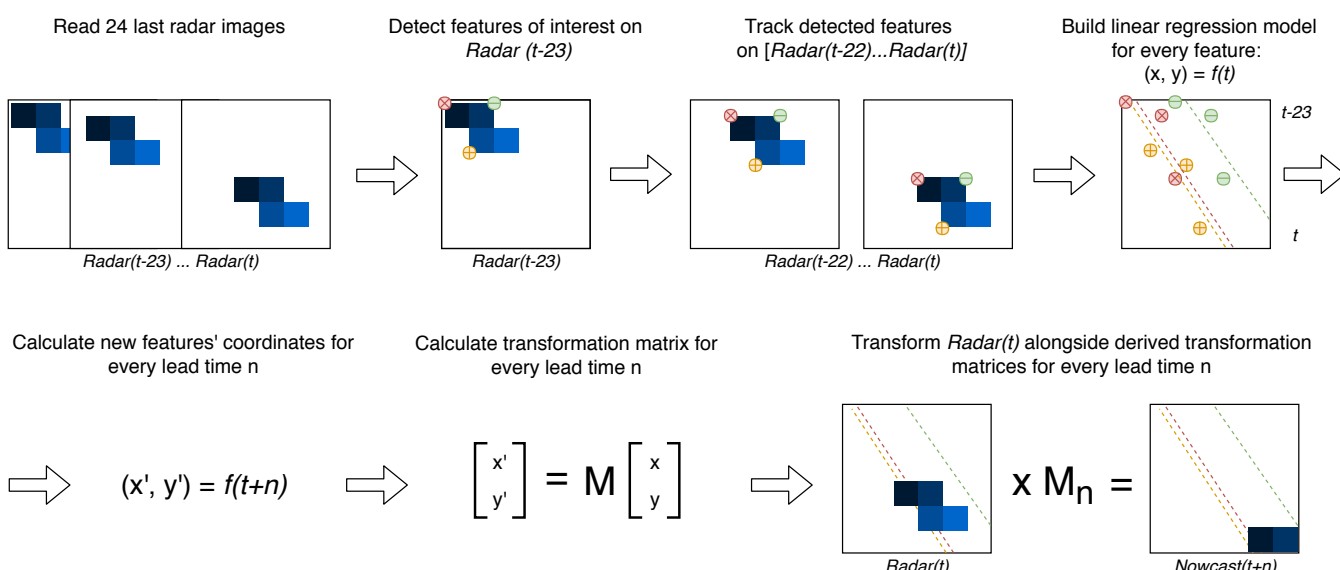

**Figure 2.** Scheme of the Sparse model

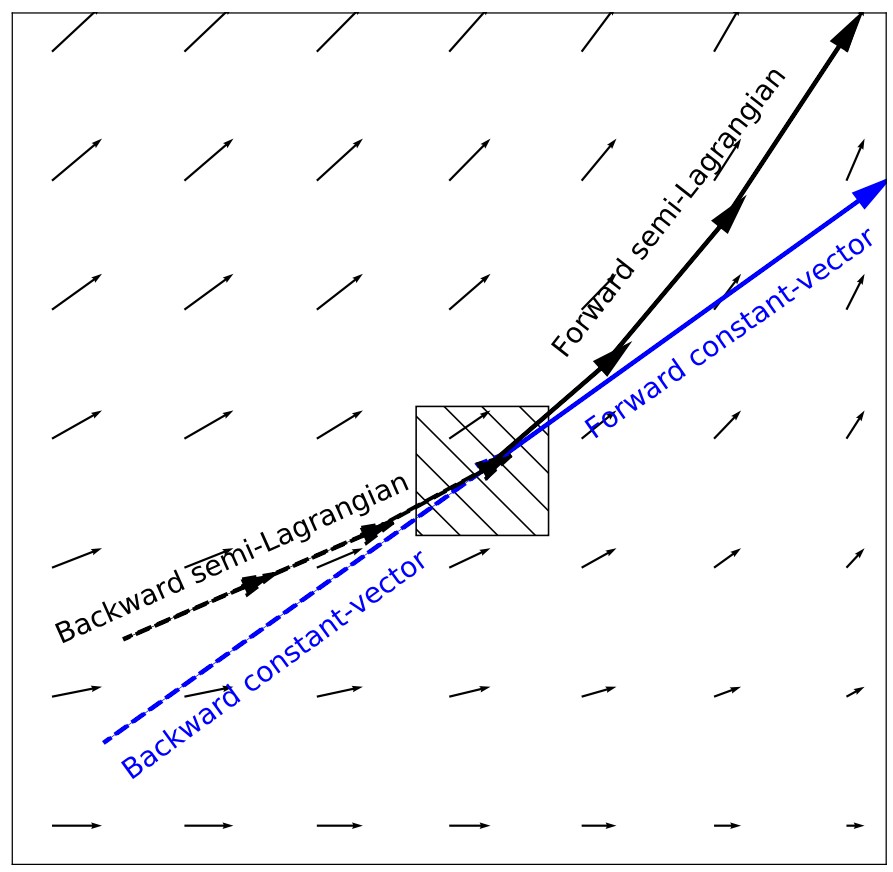

**Figure 3.** Displacement vectors of four proposed advection schemes: forward/backward constant vector and forward/backward semi-Lagrangian

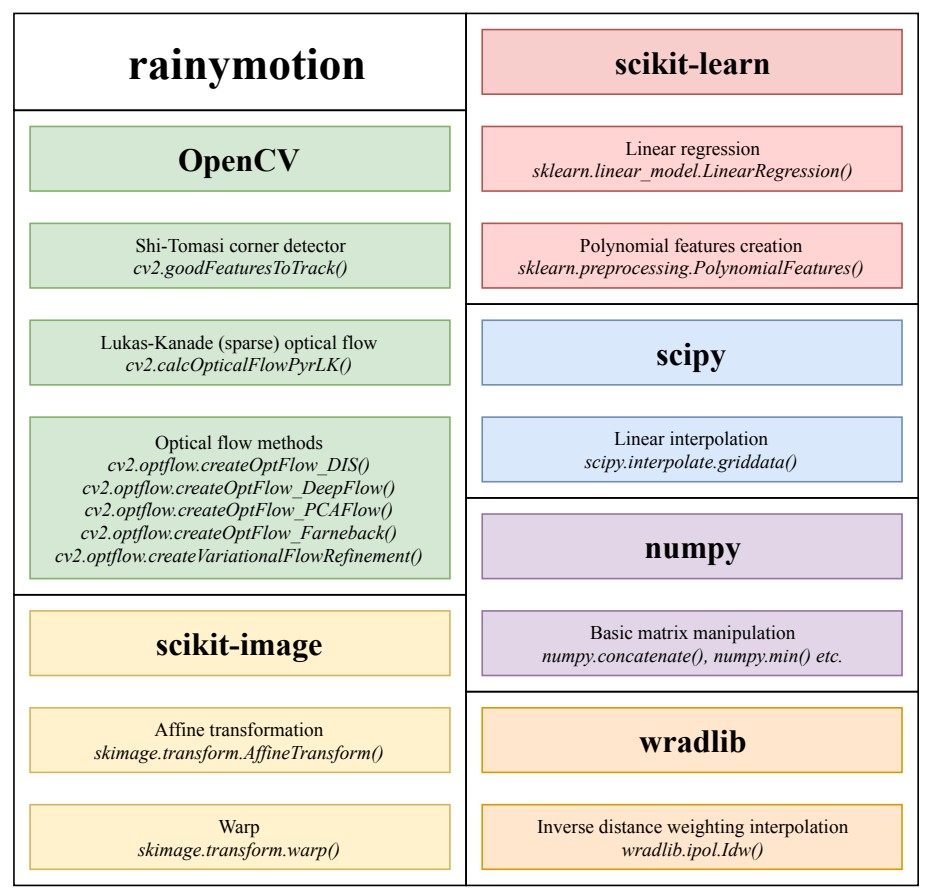

**Figure 4.** Key Python libraries for *rainymotion* library development

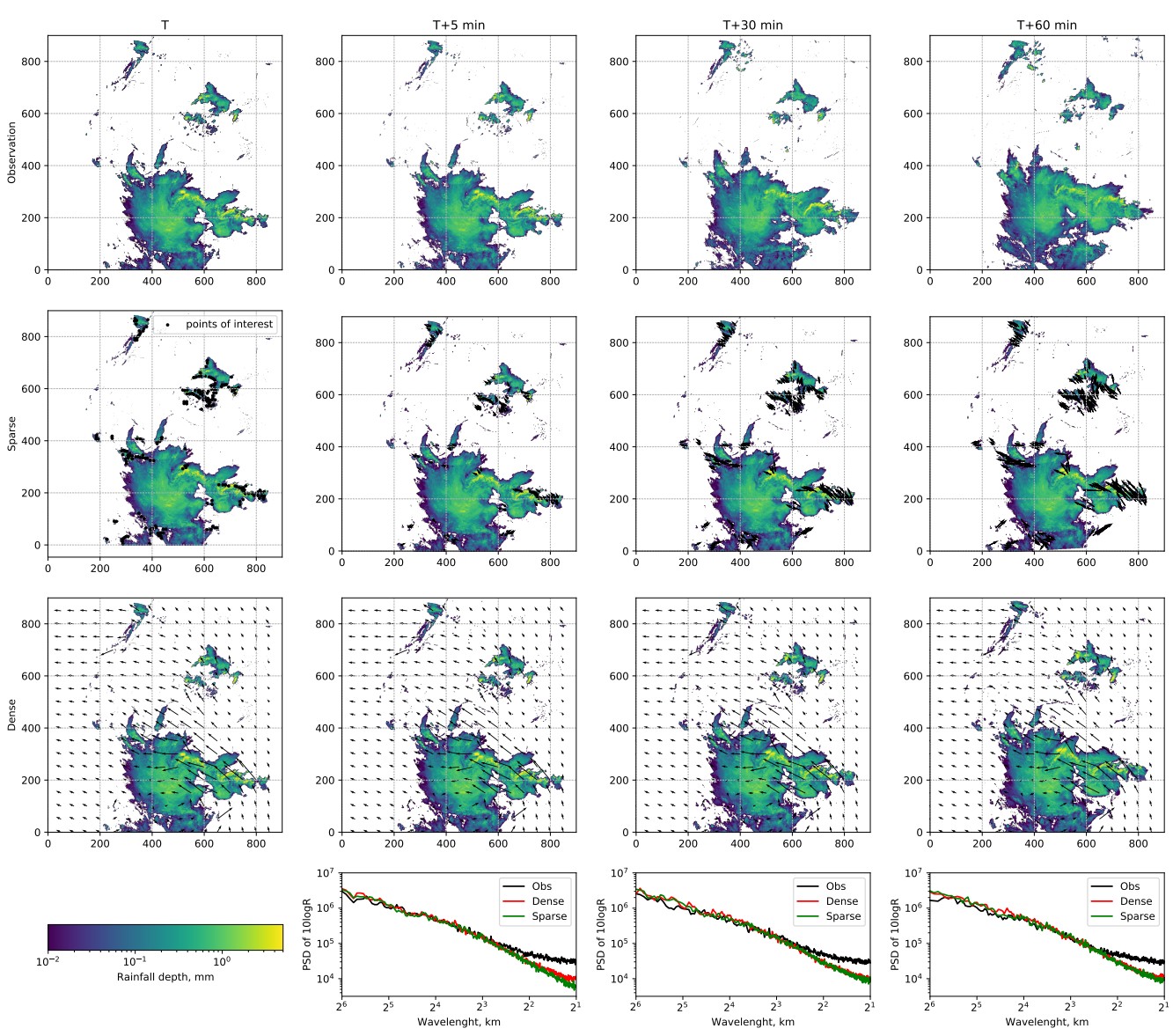

**Figure 5.** Example of the nowcasting models output (SparseSD and Dense models) for the timestep "2016-05-29 19:15" and corresponding level of numerical diffusion (the last row)

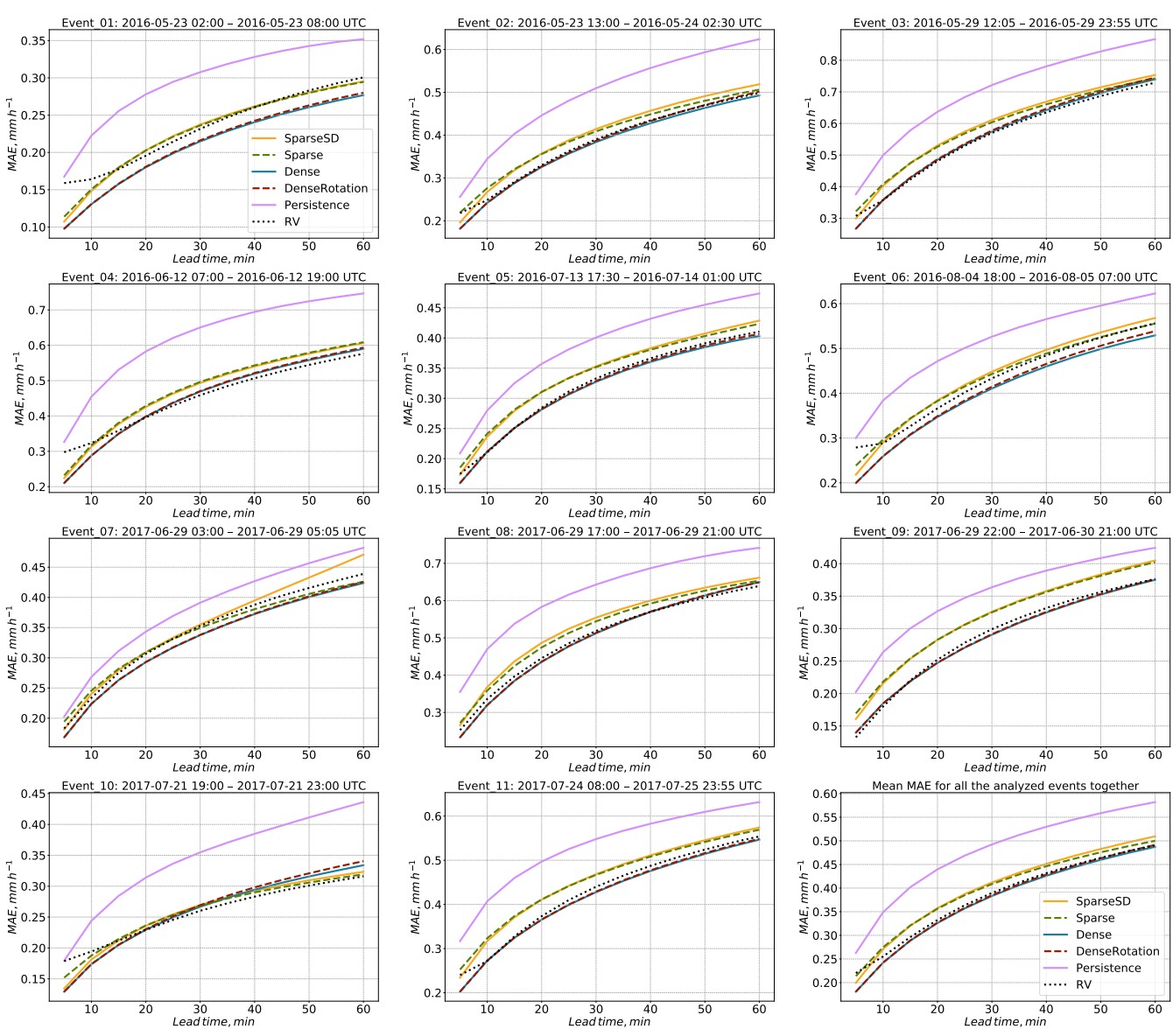

**Figure 6.** Verification of the different optical flow based nowcasts in terms of MAE for 11 precipitation events over Germany

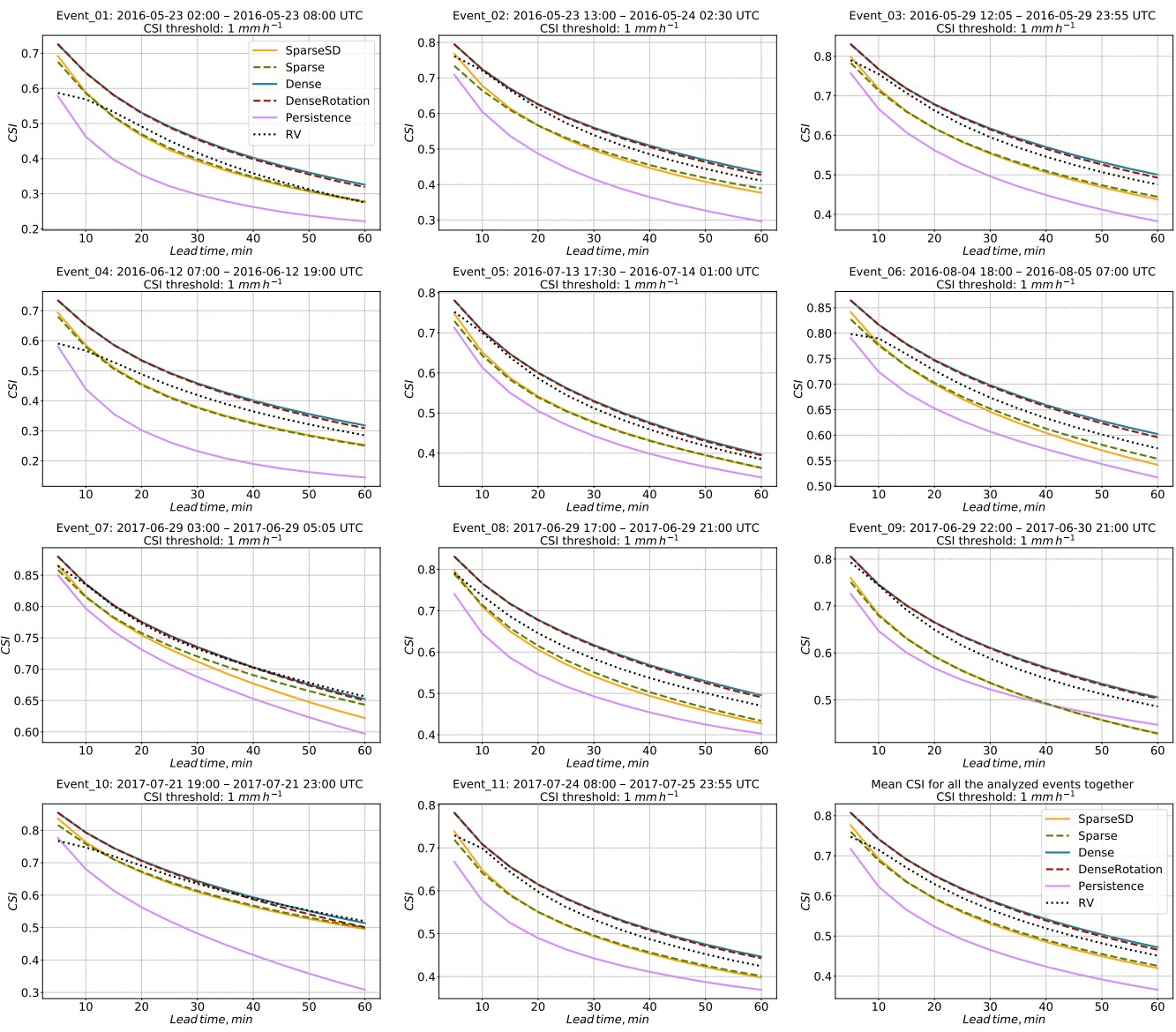

**Figure 7.** Verification of the different optical flow based nowcasts in terms of CSI for the threshold of 1 mm h$^{-1}$ for 11 precipitation events over Germany

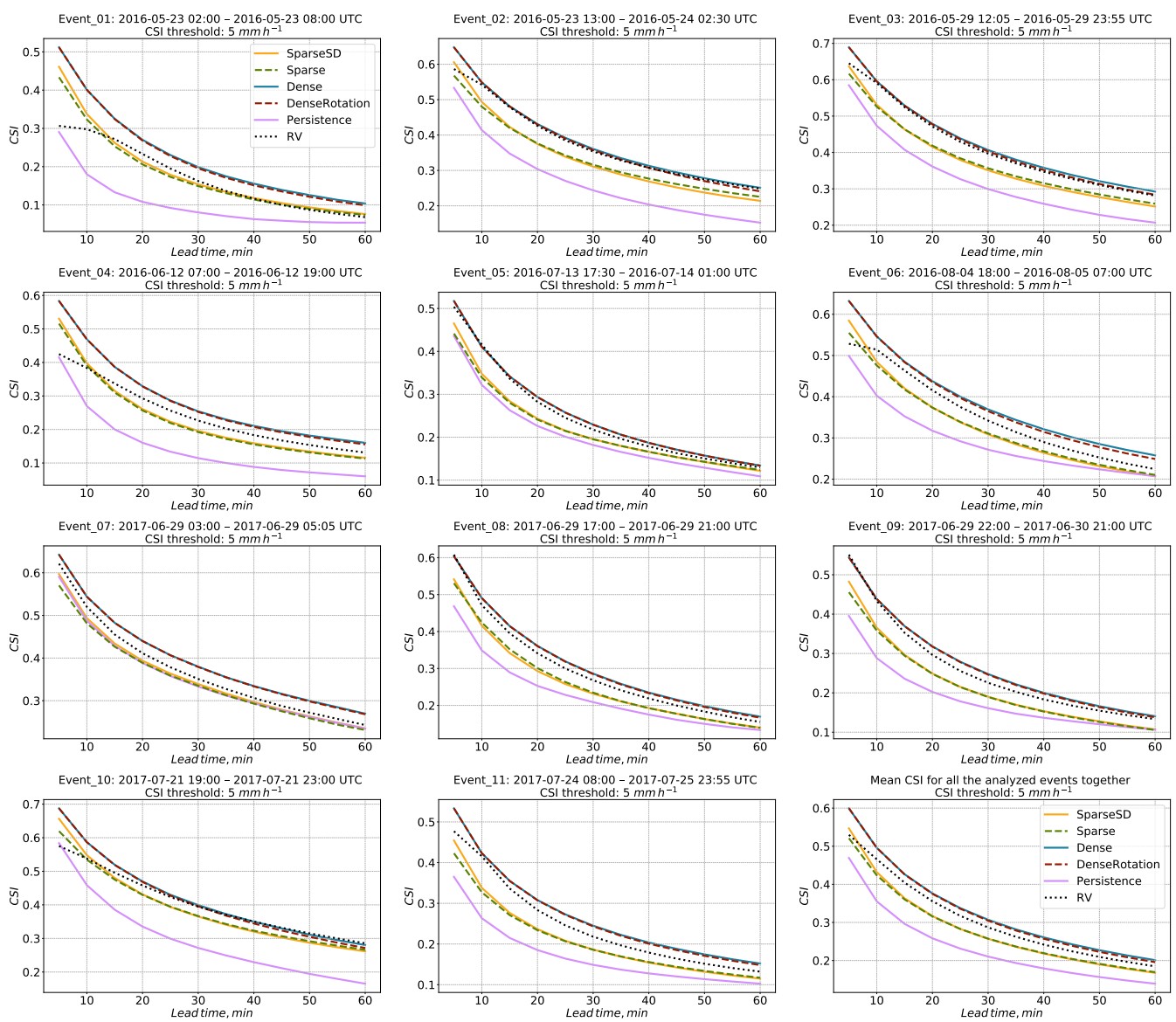

**Figure 8.** Verification of the different optical flow based nowcasts in terms of CSI for the threshold of 5 mm h$^{-1}$ for 11 precipitation events over Germany

**Table 1.** Overview of the developed nowcasting models and their computational performance. Nowcasting experiments were carried out for one hour lead time in 5 min temporal resolution (12 resulting nowcast frames in total) using the RY radar data (spatial resolution of 1 km, grid size 900×900) and a standard office PC with an Intel® Core™ i7-2600 CPU (8 cores, 3.4 GHz)

| Model name | Input radar images | Default tracking algorithm | Extrapolation | Computational time (tracking/ extrapolation/ total), s |
|---|---|---|---|---|
| SparseSD | 2 | Shi–Tomasi corner detector, Lucas–Kanade optical flow | Constant delta-change, affine warping | 0.2 / 1.1 / 1.3 |
| Sparse | 3-24 | Shi–Tomasi corner detector, Lucas–Kanade optical flow | Linear regression, affine warping | 0.1 / 1.1 / 1.2 |
| Dense | 2 | DIS optical flow | Backward constant-vector advection scheme | 0.2 / 5.5 / 5.7 |
| DenseRotation | 2 | DIS optical flow | Backward semi-Lagrangian advection scheme | 3.2 / 5.7 / 8.9 |

**Table 2.** Characteristics of the selected events

| Event # | Start | End | Duration, hours | Maximum extent, km$^2$ | Extent >1 mm h$^{-1}$, % |
|---------|-------|-----|-----------------|------------------------|--------------------------|
| Event 1 | 2016-05-23 2:00 | 2016-05-23 8:00 | 6 | 159318 | 42 |
| Event 2 | 2016-05-23 13:00 | 2016-05-24 2:30 | 13.5 | 135272 | 56 |
| Event 3 | 2016-05-29 12:05 | 2016-05-29 23:55 | 12 | 160095 | 72 |
| Event 4 | 2016-06-12 7:00 | 2016-06-12 19:00 | 12 | 150416 | 53 |
| Event 5 | 2016-07-13 17:30 | 2016-07-14 1:00 | 7.5 | 145501 | 62 |
| Event 6 | 2016-08-04 18:00 | 2016-08-05 7:00 | 13 | 168407 | 74 |
| Event 7 | 2017-06-29 3:00 | 2017-06-29 5:05 | 2 | 140021 | 70 |
| Event 8 | 2017-06-29 17:00 | 2017-06-29 21:00 | 4 | 182561 | 60 |
| Event 9 | 2017-06-29 22:00 | 2017-06-30 21:00 | 23 | 160822 | 75 |
| Event 10 | 2017-07-21 19:00 | 2017-07-21 23:00 | 4 | 63698 | 77 |
| Event 11 | 2017-07-24 8:00 | 2017-07-25 23:55 | 16 | 253666 | 63 |

**Table 3.** Mean model metrics for different lead time periods

| Model | Lead time (from–to), min | |
|---|---|---|
| | 5–30 | 35–60 |
| MAE, mm h$^{-1}$ | | |
| Dense | 0.30 | 0.45 |
| DenseRotation | 0.30 | 0.45 |
| RV | 0.31 | 0.45 |
| CSI, threshold=0.125 mm h$^{-1}$ | | |
| Dense | 0.78 | 0.64 |
| DenseRotation | 0.78 | 0.64 |
| RV | 0.76 | 0.61 |
| CSI, threshold=0.25 mm h$^{-1}$ | | |
| Dense | 0.76 | 0.61 |
| DenseRotation | 0.76 | 0.61 |
| RV | 0.74 | 0.59 |
| CSI, threshold=0.5 mm h$^{-1}$ | | |
| Dense | 0.73 | 0.57 |
| DenseRotation | 0.73 | 0.57 |
| RV | 0.70 | 0.55 |
| CSI, threshold=1 mm h$^{-1}$ | | |
| Dense | 0.68 | 0.52 |
| DenseRotation | 0.68 | 0.51 |
| RV | 0.65 | 0.49 |
| CSI, threshold=5 mm h$^{-1}$ | | |
| Dense | 0.42 | 0.24 |
| DenseRotation | 0.42 | 0.23 |
| RV | 0.39 | 0.22 |