# Peer review of "Optical flow models as an open benchmark for radar-based precipitation nowcasting (rainymotion v0.1)"

_Geoscientific Model Development, 2018_

## Referee Comment (RC1) · L. Foresti (Referee) · 30 Oct 2018

Review of paper:

**Optical flow models as an open benchmark**
**for radar-based precipitation nowcasting (rainymotion v0.1)**

by Georgy Ayzel, Maik Heistermann, and Tanja Winterrath

**Summary**
The authors present a new open source python library for benchmarking radar-based nowcasting systems based upon optical flow techniques. They propose 2 local (sparse) and 2 global (dense) approaches for motion field estimation and extrapolation of radar echoes. The nowcasts are verified using 11 precipitation events over Germany and compared to the operational nowcasting product of the DWD. Despite some variability of forecast accuracy, the dense group of techniques provides the best performance.

**General comments**
I am very glad to see this open source initiative in the field of radar-based precipitation nowcasting and I fully support it. I enjoyed reading the paper and invested significant time to provide a useful review. Hopefully, the authors will appreciate my efforts and suggestions on how to improve the manuscript.

Below you will find a table with the detailed comments line by line.
My main comments on the paper are summarized here:

- The forecast verification is well done, but in my opinion it should include a verification of the statistical properties of the advected rainfall fields to understand the degree of numerical diffusion, which can be a major problem in precipitation nowcasting if not properly handled. Such effect usually leads to an undesired smoothing of the precipitation fields, which reduces the more interesting high rainfall intensities and complicates the inter-comparison of models.
- As the paper presents new optical flow and advection techniques, it must include some additional figures showing examples of motion fields and precipitation nowcasts, e.g.:
  - A multi-panel figure with vector plots of the motion fields retrieved by the different methods overlaid on top of radar images (for example one "rotational" precipitation event).
  - A multi-panel figure showing examples of observed and nowcasted precipitation fields at different lead times, e.g. 30 or 60 minutes. This would be very useful to understand the quality and realism of the advected rainfall fields, and check whether there are any artefacts due to numerical diffusion and interpolation processes.
- Some statements in the literature review are a bit imprecise and could be improved.

**Specific comments**

| Page 1, line 3, Page 2, line 14 | "extrapolate the motion" -> "extrapolate the radar echoes". The motion field is usually kept fixed and only the radar echoes are extrapolated, although in some cases it may be beneficial to extrapolate the motion field together with the precipitation echoes. |
|---|---|
| Page 2, line 4 | The cited approaches (analogue, local Lagrangian and stochastic) were mentioned in the context of probabilistic precipitation nowcasting. They all provide empirical estimates of the probability density function in different ways. Please update accordingly. |

| | |
|---|---|
| Page 2, lines 5-6 | Foresti et al. (2015) did not use the correlation coefficient as a measure of similarity to retrieve the analogues (as done e.g. by Atencia et al., 2015), but rather the Euclidian distance in the space of principal components. Please adjust the statement. |
| Page 2, lines 8-10 | I think there is some confusion about the definition of "local Lagrangian method". The cited paper (Foresti et al., 2015) follows the definition of Germann and Zawadzki (2004), which defines the "local Lagrangian" as one possible method to derive a probabilistic nowcast. This is achieved by collecting the precipitation values upstream in a local neighbourhood, whose size is increased as a function of lead time. |
| Page 3, line 1 | I fully agree that optical flow libraries have been around for long, but they cannot be directly applied for the retrieval of radar echo motion without important adaptations and tests. For example, they must be tuned to represent the typical range of advection speeds of real precipitation fields, they must be spatially dense and extrapolate well also in regions without precipitation, etc. This is why papers like yours are important contributions to make the necessary adaptations and tests. |
| Page 3, line 8
Page 4, line 24 | It would be interesting to know why you decided not to include in the list of benchmark extrapolation techniques the backward-in-time semi-Lagrangian scheme, which is generally accepted to be the most appropriate method (Germann and Zawadzki, 2002). The forward scheme is known to produce holes in the precipitation field in presence of divergent vectors, which need to be interpolated. This inevitably leads to additional numerical diffusion. |
| Page 3, line 26 | I cannot understand properly why you mention the concept of scale-dependence in the context of local LK methods. Please explain how local optical flow techniques account for scale-dependence. |
| Page 4, line 5 | I am a bit worried that the use of warping and interpolation of discontinuities in the advected radar field can lead to serious numerical diffusion effects. The most appropriate method to test this issue is to compute the Fourier spectrum of the original and advected fields to check whether there is loss of power at the high spatial frequencies (see Fig. 10 in Germann and Zawadzki, 2002). A simpler approach would be to compare the histogram of nowcasted rainfall fields at different lead times with the one of the last observed radar image. The variance and histogram should be conserved during the extrapolation. |
| Page 4, line 26 | I agree that the constant-vector approach does not explicitly allow to account for rotation. However, if the advection is applied recursively in short time steps the rotation can be approximated by a set of short straight lines (at the cost of stronger diffusion). Despite this fact, I believe that a good implementation of the semi-Lagrangian scheme should consistently give better (or comparable) results than the constant-vector approach. |

| Page 4, line 29 | Also here I would study the effect of numerical diffusion caused by the interpolation. Numerical diffusion can also have undesired consequences when comparing (benchmarking) different nowcasting models. In fact, a precipitation nowcast that loses power at the high spatial frequencies will be generally smoother. This behavior will be rewarded in terms of some verification scores (in particular the MAE/RMSE), which affects the comparison with other models. A fair comparison of different nowcast systems should be done at similar spatial scales, for example using Fourier or wavelet decompositions. |
|---|---|
| Page 6, line 10 | "programmatic realization" is a strange expression. |
| Page 6, line 31 | "rainfall depth product". Is it the instantaneous intensity in mm/hr or an accumulation? |
| Page 6, line 23 | It would be very interesting to move the CSI verification at a threshold of 5 mm/hr from the supplementary material to the actual paper. These rainrates are the ones that are relevant to trigger warnings for severe weather. |
| Page 8, lines 5-10 | You are correct. Detailed motion fields provide better skill at short lead times, while smoother motion fields are more adapted for longer lead times. Similarly to precipitation fields, the motion fields also have an intrinsic predictability (persistence). This can be exploited by gradually smoothing the motion field in a way that is consistent with its predictability. |
| Page 8, line 14 | All the proposed solutions to the problem of low predictability at convective scales are based on the optical flow and are all valid options. However, precipitation, and in particular the one of convective nature, has a large unpredictable component that we will likely never be able to predict. Therefore, the nowcasting community needs to admit the incapability of providing accurate deterministic precipitation forecasts and find ways to estimate and communicate the inherent uncertainty. I am glad that you presented this issue in the conclusion at page 9, lines 22-25, but it would be a good idea to make this point stronger. |
| Page 9, line 20-21 | I also believe that we should not discard the Sparse models. One possibility is to make them "dense" by interpolating the motion vectors before applying the advection scheme (hopefully semi-Lagrangian). |
| Figures 1 and 2 | These are extremely clean and nice presentations of the methods. |
| Figure 3 | You may add in the caption that the figure shows the forward-in-time semi-Lagrangian method. |
| Figure 5 | You may consider writing a more descriptive figure caption, e.g. "Verification of the different optical flow based nowcasts in terms of MAE for 11 precipitation events over Germany". |
| Page 7 line 15, Figures 6-7 | Is there an explanation on why the RADVOR nowcasting method performs poorly in the first 5-10 minutes? The effect seems quite systematic and I have a |

| | |
|---|---|
| | hard time explaining it with the faster movement of precipitation fields. |
| Page 9, lines 27-30 | With respect to the use of open source libraries to promote the developments in the field of nowcasting, you could also mention how you would imagine the contribution from rainymotion to the developments of other projects, as for example the probabilistic nowcasting library pysteps (https://pysteps.github.io/). In my opinion, any improvement in optical flow methods, e.g. using the rainymotion library, will also have a positive impact on the quality of probabilistic nowcasts. This could represent an interesting synergy between the two libraries, in line with the open source philosophy. |

**References**

- Atencia, A., and I. Zawadzki, 2015: A comparison of two techniques for generating nowcasting ensembles. Part II: Analogs selection and comparison of techniques. Mon. Wea. Rev., 143 (7), 2890–2908.
- Germann, U. and I. Zawadzki, 2002: Scale-Dependence of the Predictability of Precipitation from Continental Radar Images. Part I: Description of the Methodology. Mon. Wea. Rev., 130, 2859–2873.
- Germann, U. and I. Zawadzki, 2004: Scale Dependence of the Predictability of Precipitation from Continental Radar Images. Part II: Probability Forecasts. J. Appl. Meteor., 43, 74–89.

---

## Referee Comment (RC2) · S. Pulkkinen (Referee) · 13 Nov 2018

**Review of "Optical flow models as an open benchmark for radar-based precipitation nowcasting (rainymotion v0.1)"**

Georgy Ayzel et al.

**Summary**

The manuscript describes a Python-based open-source software package for radar-based precipitation nowcasting. Verification of the methods is done by using the DWD radar data, and an operational nowcasting product is used as the baseline method. To my knowledge, there is not any existing open source nowcasting library that is documented in the form of a scientific publication. Therefore, the paper makes a great contribution to the field. What also increases the value of the work is that the optical flow and extrapolation methods are based on ideas that have not been traditionally used in the field of precipitation nowcasting (i.e. using sparse feature extraction and forward extrapolation). However, I have some suggestions to improve the presentation of the work.

**Relation to previous work and literature review**

- There are two important classes of optical flow methods that are only briefly mentioned or not mentioned at all:

  - In the variational methods, a smoothness constraint is added to the optical flow equations and they are solved "globally" over the whole domain. The key practical difference to the "local" methods, such as Farneback and Lucas-Kanade is that the motion field is automatically filled to areas of no precipitation.

  - In the spectral methods, the Fourier transform is applied to the inputs and the optical flow equations are solved in the spectral domain. The authors could add a citation to [3].

- There are several widely used optical flow algorithms developed in the machine vision literature. The authors could cite the Brox and CLG algorithms ([1] and [2]). These have also publicly available C implementations (see the IPOL journal).

- The paper cites to a large number of references where more advanced probabilistic nowcasting methods are described. Therefore, in the third paragraph of Section 6 the authors should be more concrete about future plans to include such features into rainymotion, and not just present ideas of potential improvements. Or will rainymotion be restricted only to deterministic extrapolation nowcasting based on Lagrangian persistence?

**Methodology**

- Precisely speaking the Farnebäck optical flow algorithm is not global or dense and should not be called such. This misuse of terminology originates from the OpenCV library.

  - The Farnebäck method is dense only in the sense that it produces gridded output instead of motion vectors for sparse feature points as Lucas-Kanade does. If you look at the paper of Farnebäck, the method is formulated as local feature matching, where the solution of the optical flow equations is done by using a polynomial approximation. As

a result, the method produces zero motion velocities to areas of no precipitation. You can verify this by plotting motion fields produced by the Farnebäck method.

- It follows from the above that when a pixel is advected into area of no precipitation and a new motion vector is taken at that location (as in the DenseRotation method), it's motion to stops at the boundary. This could explain why Dense has in many cases better performance than DenseRotation.

- In Germann and Zawadzki (2002), the authors conclude that the backward semi-Lagrangian has better performance than the forward method. In fact, a majority of existing nowcasting methods use the former that is widely regarded as the best approach. However, here the authors use only the latter. If possible, the authors could also implement the backward method and include it in the performance comparison.

- Using the backward method would require filling the gaps in the motion field on areas of no precipitation. Otherwise, no precipitation would be advected into areas where it does not exist at the nowcast start time. A simple distance-weighted interpolation should be sufficient for this purpose. For the above reason, using gap-filling would also improve the performance of the forward semi-Lagrangian method.

- Note that the gap-filling is automatically done in the variational methods without the need for separate post-processing of the motion field. Therefore, such methods are truly dense and global. The authors could consider implementing a variational method and include it in the performance comparison.

**The software library**

- Sections 2.4 and 3: Is the library restricted only to using the DWD data? Please add discussion about how to use the library with other file formats? For instance, by using wradlib this should be easily done because it supports a large number of different formats.

**Verification**

- Section 2.6: MAE could be computed conditionally over those pixels where both the nowcast and the verifying observation exceed the detection threshold. Otherwise, there would be overlap with the CSI statistic as both penalize incorrect forecasts of precipitation/no precipitation.

- A large number of CSI and MAE statistics are shown for different lead times. There could be more analysis of the results.

  - There is no indication about what can be considered as a good CSI or MAE value for the nowcast to be usable. Can you give some thresholds?

  - The differences between the methods (excluding Persistence) are relatively small in terms of CSI and MAE statistics. Based on such differences, the authors should be more careful when claiming that some method is better than another. For instance the maximum mean difference between Dense and DenseRotation is only 0.01 according to Table 3.

- Figures 5-7 and p. 9, lines 19-21. The authors should indeed take a closer look on why the performance of the sparse methods is poor. Some comments about this:

  - The relevant parameter here is the number of features used in the tracking and nowcasting. If this number is too small, the motion vectors of the features are not representative of the large-scale motion field. Can you check this by adjusting the thresholds in the feature detector?

  - In addition, can you specify somewhere how many feature points are used with the sparse methods because this is a key parameter?

  - Another point missed in the paper is that the corner detector tends to pick features that have high intensitites and gradients. Therefore, a very careful quality control is needed to ensure that the features are precipitation and not some random artefacts in the radar data. Can you be sure that the quality control is sufficient?

  - Even if the features are precipitation, they represent small-scale phenomena that can have very different motion from the large-scale advection field. Thus, the representativity of such features can be very poor.

- Forecasting the occurrence of precipitation/no precipitation for high intensities is highly relevant for practical applications. Therefore, I would suggest moving the results with the 5 mm/h threshold from the supplementary material to the main paper.

**Figures**

- Since the motion field determination plays a key role in the paper, the authors should show at least one figure with an observed precipitation field and the computed motion field plotted on the same figure. Even better would be a figure showing motion vectors of features and motion fields computed by using different methods.

- Figure 4: Are names of individual functions relevant here? Consider removing them.

**Minor details**

- p.4, lines 3-6 and Figure 1: How exactly is the affine transformation matrix calculated. In particular, is a single matrix estimated for all features or is this done separately for each feature?

- p.5, line 24: Why the HDF5 file format was chosen? Please add some justification for this.

- p.9, lines 7-9: I don't understand what this means. Can you clarify?

- p.9, line 24: stochastic accounting <- stochastic modeling?

**References**

[1] T. Brox, A. Bruhn, N. Papenberg and J. Weickert, High Accuracy Optical Flow Estimation Based on a Theory for Warping, ECCV 2004: 8th European Conference on Computer Vision, Prague, Czech Republic, May 11-14, 2004. Proceedings, Part IV, 25-35, 2004.

[2] A. Bruhn, J. Weickert and C. Schnörr, Lucas/Kanade Meets Horn/Schunck: Combining Local and Global Optic Flow Methods, International Journal of Computer Vision, 61(3), 211-231, 2005.

[3] E. Ruzanski, V. Chandrasekar and Y. Wang, The CASA Nowcasting System, Journal of Atmospheric and Oceanic Technology, 28(5), 640-655, 2011.

Image Processing On Line (IPOL):

http://www.ipol.im/pub/art/2013/21

http://www.ipol.im/pub/art/2015/44

---

## Referee Comment (RC3) · R. Uijlenhoet (Referee) · 15 Nov 2018

I would like to congratulate Maik Heistermann and colleagues with yet another valuable contribution to the rapidly developing field of open software, in this case in the weather radar domain. They have previously pioneered the open weather radar processing software through their python software library WRADLIB. Now they're doing the same regarding radar-based rainfall nowcasting. The timing couldn't be better, given the growing interest for this subject on all continents.

Overall, I am very happy with this manuscript and recommend to accept it with minor revisions. I have the following remarks that I invite the authors to consider taking into

account:

- References to important papers from Marc Berenguer, Daniel Sempere-Torres and Geoff Pegram are missing (SBMcast, etc.). These are very relevant papers in the context of this manuscript, which discuss the issue of spectral decomposition of precipitation fields and scale-dependent radar nowcasting.

- Reference to Berne et al. (2004; JoH) is missing. This is a (by now) classical paper on space-time scales of rainfall fields required for (urban) hydrological applications.

- Reference to pySTEPS appears to be missing (https://github.com/pySTEPS). This is the open source Python version of STEPS. Highly relevant given the topic and focus of this manuscript.

- Please provide some more detailed background information concerning: Shi–Tomasi corner detector (Shi and Tomasi, 1994); Lucas–Kanade optical flow algorithm (Lucas and Kanade, 1981); affine transformation matrix (Schneider and Eberly, 2003); warping and interpolation (Wolberg, 1990).

- "Supplementary" –> "Supplementary Information" (several times in the manuscript).

- P.4, l.8: "24 recent radar images" –> "24 most recent radar images".

- P.5, l.20: "models' description" –> "model description".

- P.6, l.25–26: "rainfall rates prediction" –> "rainfall rate prediction".

- P.8, l.6: Insert comma before "which".

- Is "RV" the same ad "RadVor"?

- General: (much) more detailed captions; figures + captions should be as self-contained as possible.

- Journal (Nature), issue, page numbers missing from reference to Bauer et al. (2015).

---

## Author Comment (AC1) · 3 Jan 2019

**Final response in the interactive discussion**

Dear Referees,

We would like to thank you for your positive comments and constructive suggestions for the improvement to our manuscript "Optical flow models as an open benchmark for radar-based precipitation nowcasting (rainymotion v0.1)". In this document, we would like to provide our responses to the comments of each of the three referees in one single document.

The referee comments turned out to be very helpful. Based on these comments, we suggest several changes to the manuscript and the rainymotion library which we will outline in detail on the following pages.

For that purpose, we will show the referee comments in **black** font, and our responses in **blue**. For the sake of clarity, we have also not reproduced some introductory parts of the referee comments in this comment. Parts that were not reproduced, are marked as [...].

We hope that the suggested changes sufficiently address the referees' concerns, so that we can, given the approval of the editor, finalize the revision of our manuscript.

Sincerely,
Georgy (on behalf of authors)

**Referee comment #1 (by Loris Foresti)**

**Main comments**

1. The forecast verification is well done, but in my opinion it should include a verification of the statistical properties of the advected rainfall fields to understand the degree of numerical diffusion, which can be a major problem in precipitation nowcasting if not properly handled. Such effect usually leads to an undesired smoothing of the precipitation fields, which reduces the more interesting high rainfall intensities and complicates the inter-comparison of models.

We entirely agree that it would be interesting to verify the statistical properties of the advected rainfall fields. It will be done as suggested in comment #11: using a periodogram of rainfall intensities of advected precipitation fields for different lead times.

2. As the paper presents new optical flow and advection techniques, it must include some additional figures showing examples of motion fields and precipitation nowcasts, e.g.:
   2.1. A multi-panel figure with vector plots of the motion fields retrieved by the different methods overlaid on top of radar images (for example one "rotational" precipitation event).
   2.2. A multi-panel figure showing examples of observed and nowcasted precipitation fields at different lead times, e.g. 30 or 60 minutes. This would be very useful to understand the quality and realism of the advected rainfall fields, and check

whether there are any artefacts due to numerical diffusion and interpolation processes.

We will add requested figures in the revised version of the manuscript.

3. Some statements in the literature review are a bit imprecise and could be improved.

We will revise the literature review in the introductory section based on several referee comments (comments #5, #6, #7, and #24 by Dr. Foresti, #1.2 and #2 by Dr. Pulkkinen, #1, #2, and #3 by Dr. Uijlenhoet).

**Specific comments**

4. Page 1, line 3, Page 2, line 14. "extrapolate the motion" -> "extrapolate the radar echoes". The motion field is usually kept fixed and only the radar echoes are extrapolated, although in some cases it may be beneficial to extrapolate the motion field together with the precipitation echoes.

We suggest to rephrase this to "[...] and then to displace the precipitation field to the imminent future (minutes to hours) based on that motion, [...]".

5. Page 2, line 4. The cited approaches (analogue, local Lagrangian and stochastic) were mentioned in the context of probabilistic precipitation nowcasting. They all provide empirical estimates of the probability density function in different ways. Please update accordingly.

Please see response to comment #7.

6. Page 2, lines 5-6. Foresti et al. (2015) did not use the correlation coefficient as a measure of similarity to retrieve the analogues (as done e.g. by Atencia et al., 2015), but rather the Euclidian distance in the space of principal components. Please adjust the statement.

Please see response to comment #7.

7. Page 2, lines 8-10. I think there is some confusion about the definition of "local Lagrangian method". The cited paper (Foresti et al., 2015) follows the definition of Germann and Zawadzki (2004), which defines the "local Lagrangian" as one possible method to derive a probabilistic nowcast. This is achieved by collecting the precipitation values upstream in a local neighbourhood, whose size is increased as a function of lead time.

Comments #5, #6, and #7 are related to one paragraph (Page 2, lines 4-9). We will rewrite the whole paragraph in accordance with the referee's suggestions and try to make the main message of this paragraph (classification of methods used for radar-based precipitation nowcasting) clearer.
We suggest to rephrase the corresponding paragraph to:
"A variety of radar-based precipitation nowcasting techniques can be classified on three major groups based on assumptions we make regarding precipitation field characteristics (Germann and Zawadski, 2002). The first group -- climatological persistence -- provides nowcasts by using

climatological values (mean or median). The second group -- Eulerian persistence -- is based on using the latest available observation as a prediction, and is thus independent from the forecast lead time. The third group -- Lagrangian persistence -- allows the extrapolation of the most recent observed precipitation field under the assumption that the motion field is persistent (Germann and Zawadzki, 2002; Woo and Wong, 2017). In addition, we can classify nowcasting methods based on introduced prediction uncertainty: In contrast to deterministic approaches, ensemble nowcast attempt to account for predictive uncertainty by including different realizations of the motion field and the evolution of rainfall intensity itself. In this study, we focus our model development around the group of Lagrangian persistence models which provide deterministic precipitation nowcasts. Yet, the unified availability of different tracking and extrapolation techniques in the rainymotion library could directly be used to construct ensembles that account for the uncertainty of rainfield displacement."

8. Page 3, line 1. I fully agree that optical flow libraries have been around for long, but they cannot be directly applied for the retrieval of radar echo motion without important adaptations and tests. For example, they must be tuned to represent the typical range of advection speeds of real precipitation fields, they must be spatially dense and extrapolate well also in regions without precipitation, etc. This is why papers like yours are important contributions to make the necessary adaptations and tests.

We agree that the original manuscript does not sufficiently address how, on the one hand, parameters of different optical flow techniques affect the specific problem of precipitation field tracking, and, on the other hand, how the results of different optical flow techniques might need further post-processing in order to enhance their usefulness for the extrapolation step (e.g. filling or interpolating zero velocities that might occur in regions of zero rainfall). Given that we also introduce further optical flow/tracking as well as extrapolation techniques (comments #9 and #12 by Dr. Foresti, #4-7 by Dr. Pulkkinen) in the revised version of the manuscript and the rainymotion library, the revised manuscript will address these requirements more precisely and comprehensively.

9. Page 3, line 8. Page 4, line 24. It would be interesting to know why you decided not to include in the list of benchmark extrapolation techniques the backward-in-time semi-Lagrangian scheme, which is generally accepted to be the most appropriate method (Germann and Zawadzki, 2002). The forward scheme is known to produce holes in the precipitation field in presence of divergent vectors, which need to be interpolated. This inevitably leads to additional numerical diffusion.

We originally implemented the forward scheme because it is more intuitive to advect the precipitation field "forward in time" and "downstream in space". Based on the referee's comment, though, we decided to complement the revised version of the rainymotion library with a backward method for the optical flow calculation.
On that basis, we repeated our benchmark experiments by using the backward scheme both for the Dense (constant-vector) and DenseRotation (semi-Lagrangian) models. Results show that the backward scheme performs slightly better for low rainfall intensity rates (under 0.5 mm/h) and longer lead times (from 30 minutes). For rainfall intensity rates over 0.5 mm/h and shorter lead times (up to 30 minutes) there are no significant differences between both schemes. Based on the new results we decided to implement the backward scheme as a default option for precipitation motion field calculation in the revised version of the rainymotion library. We will

update the revised version of the manuscript and the supplementary material in accordance with the new results.

However, we also want to note that the intercomparison of different advection schemes provided in Germann and Zawadzki (2002) cannot, in our opinion, be interpreted in a way that "backward-in-time semi-Lagrangian scheme [...] is generally accepted to be the most appropriate method". In the corresponding paper, the forward-in-time scheme is concerted with a gaussian redistribution of advected rainfall in contrast to the interpolation used for backward-in-time scheme. In our library, we adapt the same "interpolate only once" idea of Germann and Zawadzki (2002) -- regardless of the direction used for velocity field (optical flow) calculation that allows intercomparison of forward and backward schemes in a similar setting. Although the new results of our intercomparison are consistent with the referee's statement on the backward scheme being superior (for low rainfall intensities and longer lead times), in our opinion further research is needed to compare the efficiency of different implementations in detail.

10. Page 3, line 26. I cannot understand properly why you mention the concept of scale-dependence in the context of local LK methods. Please explain how local optical flow techniques account for scale-dependence.

Our intention was to highlight that for the Sparse group of rainymotion's tracking models we use distinct "corners" instead of storm cells -- this eliminates the need to specify arbitrary and scale dependent characteristics of "precipitation features" while the identification of "corners" depends only on the gradient sharpness in a cell's neighborhood. Of course this will not solve the issue of scale-dependence of the average motion itself. We will clarify these aspects in the revised manuscript.

11. Page 4, line 5. I am a bit worried that the use of warping and interpolation of discontinuities in the advected radar field can lead to serious numerical diffusion effects. The most appropriate method to test this issue is to compute the Fourier spectrum of the original and advected fields to check whether there is loss of power at the high spatial frequencies (see Fig. 10 in Germann and Zawadzki, 2002). A simpler approach would be to compare the histogram of nowcasted rainfall fields at different lead times with the one of the last observed radar image. The variance and histogram should be conserved during the extrapolation.

We will update the revised version of the manuscript with figures that describe the level of numerical diffusion for the different models by using the nowcasts' spectral power density for different lead times.

12. Page 4, line 26. I agree that the constant-vector approach does not explicitly allow to account for rotation. However, if the advection is applied recursively in short time steps the rotation can be approximated by a set of short straight lines (at the cost of stronger diffusion). Despite this fact, I believe that a good implementation of the semi-Lagrangian scheme should consistently give better (or comparable) results than the constant-vector approach.

We agree that an accurate implementation of the semi-Lagrangian scheme should yield a skill that is at least equivalent to the constant-vector approach. We have found two possible reasons

why our original implementation did not achieve that: 1. Errors in the estimation of motion fields (e.g. with anomalies, artefacts etc.) could affect the forecast in the semi-Lagrangian advection scheme more than in the constant-vector scheme, since displacement vectors from regions of higher uncertainty might be "activated" more frequently; 2. Higher complexity of semi-Lagrangian scheme implementation which involves interpolation on two levels: when we advect each pixel and try to find the new velocity vector for any new pixel location, and during the final interpolation of intensities.

We attempted to address the estimation of field motion using the (local) Farnebäck optical flow method by implementing a variational refinement procedure to smooth the velocity field, and to get rid of spurious velocities, and by implementing different global optical flow methods that usually provide more smooth and robust motion fields (see also comment #4.1 and #7 by Dr. Pulkkinen). As a result, we included both the variational refinement and different global methods for the tracking step in the rainymotion library, and included these approaches in our benchmarking experiments.

According to these new results, the implementation of the Dense Inverse Search (DIS) global optical flow method (Kroeger et al., 2016) provides better results than the Farnebäck method with variational refinement and other global methods such as DeepFlow (Weinzaepfel et al., 2013) and PCAFlow (Wulff and Black, 2015). Based on these new findings, we decided to use the DIS method as a default method for the precipitation motion field calculation in the revised version of the rainymotion library. We also found that using the DIS method, our results show no significant difference between Dense and DenseRotation models. That confirms the strong influence of motion field estimation on the performance of the DenseRotation model.

We will update manuscripts accordingly and show the intercomparison of of different optical flow methods in the supplementary material.

Kroeger, T., Timofte, R., Dai, D., & Van Gool, L. (2016, October). Fast optical flow using dense inverse search. In *European Conference on Computer Vision* (pp. 471-488). Springer, Cham.

Weinzaepfel, P., Revaud, J., Harchaoui, Z., & Schmid, C. (2013). DeepFlow: Large displacement optical flow with deep matching. In *Proceedings of the IEEE International Conference on Computer Vision* (pp. 1385-1392).

Wulff, J., & Black, M. J. (2015). Efficient sparse-to-dense optical flow estimation using a learned basis and layers. In *Proceedings of the IEEE Conference on Computer Vision and Pattern Recognition* (pp. 120-130).

13. Page 4, line 29. Also here I would study the effect of numerical diffusion caused by the interpolation. Numerical diffusion can also have undesired consequences when comparing (benchmarking) different nowcasting models. In fact, a precipitation nowcast that loses power at the high spatial frequencies will be generally smoother. This behavior will be rewarded in terms of some verification scores (in particular the MAE/RMSE), which affects the comparison with other models. A fair comparison of different nowcast systems should be done at similar spatial scales, for example using Fourier or wavelet decompositions.

Please see response to comment #11.

14. Page 6, line 10. "programmatic realization" is a strange expression.

We will rephrase to "In this study we used the RV product data as an operational baseline and did not reimplement the underlying algorithm itself."

15. Page 6, line 31. "rainfall depth product". Is it the instantaneous intensity in mm/hr or an accumulation?

The RY product represents rainfall depth in mm for a five minute interval which is however derived from an instantaneous intensity considered representative for that interval.

16. Page 6, line 23. It would be very interesting to move the CSI verification at a threshold of 5 mm/hr from the supplementary material to the actual paper. These rainrates are the ones that are relevant to trigger warnings for severe weather.

We will update Figure 6 to represent the CSI for the threshold of 1 mm/h and Figure 7 to represent the CSI for the threshold of 5 mm/h.

17. Page 8, lines 5-10. You are correct. Detailed motion fields provide better skill at short lead times, while smoother motion fields are more adapted for longer lead times. Similarly to precipitation fields, the motion fields also have an intrinsic predictability (persistence). This can be exploited by gradually smoothing the motion field in a way that is consistent with its predictability.

We agree. In fact, users can use the library to implement such ideas.

18. Page 8, line 14. All the proposed solutions to the problem of low predictability at convective scales are based on the optical flow and are all valid options. However, precipitation, and in particular the one of convective nature, has a large unpredictable component that we will likely never be able to predict. Therefore, the nowcasting community needs to admit the incapability of providing accurate deterministic precipitation forecasts and find ways to estimate and communicate the inherent uncertainty. I am glad that you presented this issue in the conclusion at page 9, lines 22-25, but it would be a good idea to make this point stronger.

We will try to emphasize this point in the revised version of the manuscript.

19. Page 9, line 20-21. I also believe that we should not discard the Sparse models. One possibility is to make them "dense" by interpolating the motion vectors before applying the advection scheme (hopefully semi-Lagrangian).

We thank referee for this comment. More generally, future research should analyse in more detail which steps in our Sparse model chain contribute the most uncertainty. We still think the combination of Sparse optical flow and warping is very efficient and promising, but should be understood better.

20. Figures 1 and 2. These are extremely clean and nice presentations of the methods.

We thank referee for this comment.

21. Figure 3. You may add in the caption that the figure shows the forward-in-time semi-Lagrangian method.

Figure 3 caption will be updated accordingly.

22. Figure 5. You may consider writing a more descriptive figure caption, e.g. "Verification of the different optical flow based nowcasts in terms of MAE for 11 precipitation events over Germany".

Figure 5 caption will be updated accordingly.

23. Page 7 line 15, Figures 6-7. Is there an explanation on why the RADVOR nowcasting method performs poorly in the first 5-10 minutes? The effect seems quite systematic and I have a hard time explaining it with the faster movement of precipitation fields.

We briefly described the possible reasons of this RADVOR behavior on Page 7 lines 14-15 and Page 8 lines 3-8. In our opinion, the use of smoothed displacement fields that focus on a large scale motion patterns particularly cause a loss of skill in the RV product for the first 5-10 minutes. We will update the corresponding paragraph of Section 5.1 (Model comparison) to make that point clearer.

24. Page 9, lines 27-30. With respect to the use of open source libraries to promote the developments in the field of nowcasting, you could also mention how you would imagine the contribution from rainymotion to the developments of other projects, as for example the probabilistic nowcasting library pysteps (https://pysteps.github.io/). In my opinion, any improvement in optical flow methods, e.g. using the rainymotion library, will also have a positive impact on the quality of probabilistic nowcasts. This could represent an interesting synergy between the two libraries, in line with the open source philosophy.

At the moment of paper submission (6th July), no reference to pySTEPS was known to us (first commit on GitHub from 9th July). We will update the introductory section to add a reference to pySTEPS, but we will also include the perspectives mentioned by the referee in the "Summary and conclusions".

**Referee comment #2 (by Seppo Pulkkinen)**

**Relation to previous work and literature review**

1. There are two important classes of optical flow methods that are only briefly mentioned or not mentioned at all:

   1.1. In the variational methods, a smoothness constraint is added to the optical flow equations and they are solved "globally" over the whole domain. The key practical difference to the "local" methods, such as Farnebäck and Lucas-Kanade is that the motion field is automatically filled to areas of no precipitation.

   1.2. In the spectral methods, the Fourier transform is applied to the inputs and the optical flow equations are solved in the spectral domain. The authors could add a citation to [3].

[3] E. Ruzanski, V. Chandrasekar and Y. Wang, The CASA Nowcasting System, Journal of Atmospheric and Oceanic Technology, 28(5), 640-655, 2011.

We thank referee for the clarification. We will add the corresponding methods and references to the introduction of the revised version of the manuscript, particularly since we added a variational approach and several global methods to the rainymotion library and our benchmark experiment - please see our response to comment #12 of Dr. Foresti.

2. There are several widely used optical flow algorithms developed in the machine vision literature. The authors could cite the Brox and CLG algorithms ([1] and [2]). These have also publicly available C implementations (see the IPOL journal).

[1] T. Brox, A. Bruhn, N. Papenberg and J. Weickert, High Accuracy Optical Flow Estimation Based on a Theory for Warping, ECCV 2004: 8th European Conference on Computer Vision, Prague, Czech Republic, May 11-14, 2004. Proceedings, Part IV, 25-35, 2004.
[2] A. Bruhn, J. Weickert and C. Schnörr, Lucas/Kanade Meets Horn/Schunck: Combining Local and Global Optic Flow Methods, International Journal of Computer Vision, 61(3), 211-231, 2005.

We will add those references to the paragraph where we mention only openCV as an open software library with optical flow algorithms implementation (Page 3 line 1 of the discussion paper).

3. The paper cites to a large number of references where more advanced probabilistic nowcasting methods are described. Therefore, in the third paragraph of Section 6 the authors should be more concrete about future plans to include such features into rainymotion, and not just present ideas of potential improvements. Or will rainymotion be restricted only to deterministic extrapolation nowcasting based on Lagrangian persistence?

We would like to thank the referee for this suggestion, yet we are hesitant whether more detailed perspectives on future developments should be elaborated in the paper. Based on its current design, rainymotion's focus is to track the motion of rainfields and to extrapolate future rainfall on that basis. At the same time, the rather low-level implementation easily allows for the flexibility to manipulate the displaced precipitation fields in order to represent -- stochastically or

deterministically -- the dynamics of precipitation intensity. Yet, there are no specific plans to implement such features to the rainymotion, but we will, in the revised version of the manuscript, highlight more explicitly the possibility to include such developments.

**Methodology**

4. Precisely speaking the Farnebäck optical flow algorithm is not global or dense and should not be called such. This misuse of terminology originates from the OpenCV library.

    4.1. The Farnebäck method is dense only in the sense that it produces gridded output instead of motion vectors for sparse feature points as Lucas-Kanade does. If you look at the paper of Farnebäck, the method is formulated as local feature matching, where the solution of the optical flow equations is done by using a polynomial approximation. As a result, the method produces zero motion velocities to areas of no precipitation. You can verify this by plotting motion fields produced by the Farnebäck method.

We thank the referee for pointing that misuse in terminology which we have not been aware of so far. In our paper we use the term "local" and "sparse" in the sense that these methods provide motion vectors at specific locations only. In contrast, we use "global" and "dense" for pointing out that the motion vectors are calculated for an every radar image pixel.
We will revise the paper in a way that terminology is both accurate and easy to understand. As a possible solution we propose to change "local" and "global" to "sparse" and "dense" in the revised version of the manuscript and provide a more detailed description of what we consider as "sparse" and "dense" models in the Section 2 (Model). Furthermore, we have actually added global optical flow techniques to the set of tracking models (please see our response to comment #12 of Loris Foresti), so the revised manuscript version will explicitly address the issue of global vs. local optical flow.

    4.2. It follows from the above that when a pixel is advected into area of no precipitation and a new motion vector is taken at that location (as in the DenseRotation method), it's motion to stops at the boundary. This could explain why Dense has in many cases better performance than DenseRotation.

At the time of submission of this manuscript and thus when the results for this paper had been produced, the implementation of the extrapolation algorithm of the dense optical flow models did in fact not account for the case that pixels are advected into regions of zero velocities. In the meantime, however, we have revised the algorithm so that zero velocities are discarded and replaced by interpolation, and the results will be updated accordingly. Yet, the hypothesis that the insufficient treatment of zero velocities was responsible for the Dense model outperforming the DenseRotation model could not yet be corroborated (please also refer to our response to comment #12 of Dr. Foresti).

5. In Germann and Zawadzki (2002), the authors conclude that the backward semi-Lagrangian has better performance than the forward method. In fact, a majority of existing nowcasting methods use the former that is widely regarded as the best approach. However, here the authors use only the latter. If possible, the authors could also implement the backward method and include it in the performance comparison.

Dr. Foresti in his comment #9 also raised this issue. Please refer to our answer there.

6. Using the backward method would require filling the gaps in the motion field on areas of no precipitation. Otherwise, no precipitation would be advected into areas where it does not exist at the nowcast start time. A simple distance-weighted interpolation should be sufficient for this purpose. For the above reason, using gap-filling would also improve the performance of the forward semi-Lagrangian method.

In the revised version of the rainymotion library we implemented the referee's suggestion of "[..] filling the gaps in the motion field on areas of no precipitation" by utilizing inverse distance weighted interpolation to fill zero-gaps in the motion field. However, the benefit of this implementation on the performance of the forward semi-Lagrangian method (the DenseRotation model) is not so distinct probably because of the reasons we highlighted in the response on the comment #12 from Dr. Foresti (motion field estimation errors by the Farnebäck algorithm and additional interpolation).

7. Note that the gap-filling is automatically done in the variational methods without the need for separate post-processing of the motion field. Therefore, such methods are truly dense and global. The authors could consider implementing a variational method and include it in the performance comparison.

We thank the referee for his recommendations regarding the implementation of variational optical flow models in the rainymotion library. We incorporated global optical flow methods which are available in opencv library as additional options for motion field calculation in the rainymotion library (see also our response to the comment #12 from Dr. Foresti), verified their skill for nowcasting and have to conclude that using more advanced global optical flow methods advances an efficiency of a semi-Lagrangian advection scheme. Based on the new obtained results we decided to replace the Farnebäck method by the global Dense Inverse Search (DIS, Kroeger et al., 2016) as a default tracking option. We will also update the supplementary material with intercomparison results of different optical flow methods.

Kroeger, T., Timofte, R., Dai, D., & Van Gool, L. (2016, October). Fast optical flow using dense inverse search. In *European Conference on Computer Vision* (pp. 471-488). Springer, Cham.

**Software library**

8. Sections 2.4 and 3: Is the library restricted only to using the DWD data? Please add discussion about how to use the library with other file formats? For instance, by using wradlib this should be easily done because it supports a large number of different formats.

There is no restriction in using different data formats because of rainymotion works directly with numpy arrays, and the data preprocessing routine is fully on the user-side. There is a set of available open software libraries for radar data reading and preprocessing (the list available on https://openradarscience.org/). We will add the corresponding information to the Section 2.4 (rainymotion Python library).

**Verification**

9. Section 2.6: MAE could be computed conditionally over those pixels where both the nowcast and the verifying observation exceed the detection threshold. Otherwise, there would be overlap with the CSI statistic as both penalize incorrect forecasts of precipitation/no precipitation.

In our study we decided to use MAE as a score from a continuous category and implement it directly without making specific thresholds (like we do for categorical category of verification scores). In our opinion, this admittedly arbitrary decision of using different verification score categories helps to represent a diversity of obtained results.

10. A large number of CSI and MAE statistics are shown for different lead times. There could be more analysis of the results.
    10.1. There is no indication about what can be considered as a good CSI or MAE value for the nowcast to be usable. Can you give some thresholds?

At the best of our knowledge, there is no convention regarding what to consider as "good" or "bad" for any verification metric commonly used in radar-based QPN. For our benchmarking experiment, the focus is on the differences of scores between the different models, not on their absolute values.

   10.2. The differences between the methods (excluding Persistence) are relatively small in terms of CSI and MAE statistics. Based on such differences, the authors should be more careful when claiming that some method is better than another. For instance the maximum mean difference between Dense and DenseRotation is only 0.01 according to Table 3.

Table 3 represents statistics which are averaged over all the analyzed events and two lead time periods (5--30, and 35--60) and primarily highlight the difference between the Dense group of rainymotion models (Dense and DenseRotation) and and the RV product (as mentioned on the Page 7, lines 27--31) -- which is more distinct than the difference between Dense and DenseRotation models themselves. For the verification procedure we also carried out the Student's independent two-sample *t*-test to find whether differences between mean CSI and MAE values for the specific lead times are significant or not (not shown in the manuscript). We found that the results of the visual inspection of the verification plots are well consistent with the formal statistical evaluation: if there is a clear difference in the plots, it is typically significant in a statistical sense.
We will update Table 3 with the new results and adjust our statements about considering one model better/worse than another correspondingly.

11. Figures 5-7 and p. 9, lines 19-21. The authors should indeed take a closer look on why the performance of the sparse methods is poor. Some comments about this:
    11.1. The relevant parameter here is the number of features used in the tracking and nowcasting. If this number is too small, the motion vectors of the features are not representative of the large-scale motion field. Can you check this by adjusting the thresholds in the feature detector?

11.2. In addition, can you specify somewhere how many feature points are used with the sparse methods because this is a key parameter?

11.3. Another point missed in the paper is that the corner detector tends to pick features that have high intensities and gradients. Therefore, a very careful quality control is needed to ensure that the features are precipitation and not some random artefacts in the radar data. Can you be sure that the quality control is sufficient?

11.4. Even if the features are precipitation, they represent small-scale phenomena that can have very different motion from the large-scale advection field. Thus, the representativity of such features can be very poor.

We agree with the referee that the sensitivity of the Sparse group of models to specific key parameters needs to be investigated more closely. Yet, we consider such an analysis beyond the scope of this study. Another study is underway that specifically and systematically focuses on the error of the forecast location of detected features based on a vast set of tracking and extrapolation techniques, and including different parameterisations as mentioned by the referee (such as the maximum number of features detected, or different approaches to filter spurious or non-representative velocities at small spatiotemporal scales). In the present manuscript under discussion, however, our aim is to present two basic and open architectures of nowcasting models based on optical flow which can serve as a baseline for future developments - as part of the rainymotion library itself or in combination with the library, and to demonstrate that these are skillful. Still, the parameters of the Shi-Tomasi corner detector provide us a possibility to control the maximum number of features, their quality (which is based on the minimal eigenvalue) and a minimum euclidean distance between the nearest identified points. A calibration of these parameters had been performed on different events and the most robust values had been set up as default parameters as follows: maximum number of features -- 200; quality level -- 0.2 (the corners with the quality measure less than the product of quality level and minimal eigenvalue will be rejected); minimum euclidean distance -- 7 pixels (the corners which have stronger neighbors in a neighborhood less than 7 pixels will be rejected). As for quality control of the actual radar data, we rely on the DWD's processing workflow that produces the RY product and which eliminates vast parts of spurious echoes. Yet, even in the presence of residual static or dynamic clutter, the tracking algorithm has proven to be robust against producing zero velocities.

12. Forecasting the occurrence of precipitation/no precipitation for high intensities is highly relevant for practical applications. Therefore, I would suggest moving the results with the 5 mm/h threshold from the supplementary material to the main paper.

We support referee's recommendation (see also comment #16 by Loris Foresti) and will transfer the corresponding figure from the supplementary to the main paper.

**Figures**

13. Since the motion field determination plays a key role in the paper, the authors should show at least one figure with an observed precipitation field and the computed motion field plotted on the same figure. Even better would be a figure showing motion vectors of features and motion fields computed by using different methods.

We agree with the referees' recommendation (see also comment #2 by Loris Foresti) and will add the requested figures to the revised version of the manuscript.

14.     Figure 4: Are names of individual functions relevant here? Consider removing them.

In our opinion, it is informative to show the key functions that we used from various libraries in order to put together the main functionality of rainymotion. It illustrates that the combination is, from a technical perspective, not too complex.

**Minor details**

15.     Page 4, lines 3-6 and Figure 1. How exactly is the affine transformation matrix calculated. In particular, is a single matrix estimated for all features or is this done separately for each feature?

The transformation matrix is calculated on the basis of all identified features. We will add this clarification to Section 2.1 (Local optical flow models).

16.     Page 5, line 24. Why the HDF5 file format was chosen? Please add some justification for this.

For all internal projects we use HDF5 database and corresponding file format as an efficient data storage with powerful set of archiving options (i.e. compression rate, chunk size) instead of using default binary files provided by the DWD. However, we propose to remove the reference to HDF5 file format and h5py library because of it is neither integral part of our analysis, nor the rainymotion library, but just a subjective choice we made regarding our research workflow. We will update the Section 3 correspondingly.

17.     Page 9, lines 7-9. I don't understand what this means. Can you clarify?

The statement "It might also be considered to combine the warping procedure for the extrapolation step with the Dense optical flow procedure for the tracking step in order to dramatically enhance computational performance" describes the idea to detect corners, then predict the future locations of these corners using the motion field from dense optical flow, and then construct the Affine Transformation Matrix for the warping based on the corner locations at forecast time and lead time $t_n$. That way, we would combine the robustness of the dense optical flow technique with the computational efficiency of the warping technique. We will clarify that idea in the revised manuscript.

18.     Page 9, line 24. Stochastic accounting <- stochastic modeling?

We thank referee for pointing out that mistake which will be corrected in the revised version of the manuscript.

**Referee comment #3 (by Remko Uijlenhoet)**

1. References to important papers from Marc Berenguer, Daniel Sempere-Torres and Geoff Pegram are missing (SBMcast, etc.). These are very relevant papers in the context of this manuscript, which discuss the issue of spectral decomposition of precipitation fields and scale-dependent radar nowcasting.
2. Reference to Berne et al. (2004; JoH) is missing. This is a (by now) classical paper on space-time scales of rainfall fields required for (urban) hydrological applications.
3. Reference to pySTEPS appears to be missing (https://github.com/pySTEPS). This is the open source Python version of STEPS. Highly relevant given the topic and focus of this manuscript.

We will include the suggested references in the introductory section. As for the missing reference to pySTEPS, we refer to our response to comment #24 of Loris Foresti.

4. Please provide some more detailed background information concerning: Shi–Tomasi corner detector (Shi and Tomasi, 1994); Lucas–Kanade optical flow algorithm (Lucas and Kanade, 1981); affine transformation matrix (Schneider and Eberly, 2003); warping and interpolation (Wolberg, 1990).

We will try to illustrate in more detail the main features of these techniques in the revised version of the manuscript.

5. Corrections, grammar and typos
   5.1. "Supplementary" –> "Supplementary Information" (several times in the manuscript).
   5.2. P.4, l.8: "24 recent radar images" –> "24 most recent radar images".
   5.3. P.5, l.20: "models' description" –> "model description".
   5.4. P.6, l.25–26: "rainfall rates prediction" –> "rainfall rate prediction".
   5.5. P.8, l.6: Insert comma before "which".

Will be fixed.

6. Is "RV" the same ad "RadVor"?

RADVOR is the entire nowcasting workflow used by the DWD. RV is a main product along that processing chain which is the forecast precipitation depth in five minute intervals over a lead time of two hours. The official main product of RADVOR, though, is the RQ product which is the precipitation depth accumulated over an interval of one hour for a lead time of two hours. It is basically obtained from the RV product, but includes an additional adjustment of the distribution function. In summary, the RV product is the part of DWD's nowcasting chain that is best comparable to our nowcasting products and the best "end product" that is available at an interval of five minutes.

7. General: (much) more detailed captions; figures + captions should be as self-contained as possible.

We will update the figure captions to make them more self-contained.

8.  Journal (Nature), issue, page numbers missing from reference to Bauer et al. (2015).

We will update the corresponding reference to the Bauer et al. (2015) paper as following:
Bauer, P., Thorpe, A., Brunet G.: The quiet revolution of numerical weather prediction, Nature, 525, 47–55, https://doi.org/10.1038/nature14956, https://www.nature.com/articles/nature14956, 2015.

---

## Author Response (AR1)

**Response letter**

Dear Referees, dear Editor,

We would like to thank you again for your positive comments and constructive suggestions for the improvement of our manuscript. In this document, we would like to provide our responses to the comments of each of the three referees including the decision on specific changes in the manuscript. For that purpose, we use the following color code:

black: original referee comment blue: our original response in the Interactive discussion green: our final response and the specific changes made in the revised manuscript

Before addressing each referee comment, we also provide a summary of the most important changes to both the manuscript and the rainymotion library in the course of this revision.

Addressing the referee comments has, in our opinion, substantially improved the paper, and we hope that the quality of the paper now allows for publication in GMD.

Sincerely, Georgy (on behalf of the authors)

**Summary of major changes to the manuscript and the rainymotion library**

**Benchmarking of the new global optical flow techniques**

Based on the reviewers' comments (comments #9 and #12 by Dr. Foresti, #4-7 by Dr. Pulkkinen), we complemented the previously used Farnebäck local optical flow algorithm with procedures to replace zero velocities and a further variational refinement of the obtained velocity field, and also included a set of global optical flow algorithms, such as DIS, DeepFlow, and PCAFlow. We updated the rainymotion library accordingly, and conducted an extensive benchmark experiment to evaluate efficiency of these different optical flow algorithms for precipitation nowcasting. Results showed that the DIS optical flow algorithm provides better results both in terms of verification metrics and computational performance. Thus, we selected the DIS optical flow algorithm as the default option for the tracking step in the Dense group of models in the rainymotion library.

**Benchmarking of different advection approaches**

Based on the reviewers' comments (comment #9 by Dr. Foresti, and #5 by Dr. Pulkkinen), we performed a benchmark experiment to verify the performance two different implementations (namely forward and backward) of the constant-vector and the semi-Lagrangian advection schemes. Results showed that the backward scheme performs *slightly* better for low rainfall intensity rates (under 0.5 mm/h) and longer lead times (from 30 minutes). For rainfall intensities above 0.5 mm/h and shorter lead times (up to 30 minutes) there are no significant differences between both schemes. Thus, we decided to implement the backward scheme as the default option for precipitation advection in the revised version of the rainymotion library.

**Increasing computational performance by a new interpolation approach**

The linear interpolation of the advected rainfall pixels to the original radar grid was a serious bottleneck in the computational performance of the Dense group of the rainymotion models for large grids (900x900 pixels in the case of the RY data). In the revised version of the rainymotion library, we replaced the linear interpolation (implemented via scipy.interpolate module) by inverse distance weighting interpolation (implemented via the wradlib.ipol module). This led to a substantial improvement in computational efficiency (by a factor of about 15), without a drop in verification efficiency.

**Exemplary investigation of effects of numerical diffusion**

Based on the comments #1, #8, #11, #12, and #13 by Dr. Foresti, we introduced an exemplary analysis of the effects of numerical diffusion in Figure 5 of the revised manuscript, based on the loss of power spectral density as compared to the observations. For that case study, we did not find any substantial loss of power at small spatial scales - at least for the lead times of up to one hour investigated in our study. We hypothesize that this encouraging result is due to the fact that we interpolate only *once per lead time* (in the Dense group of models, for both the forward and the backward scheme), and that the warping procedure intrinsically conserves power at small scales while interpolation effects are negligible (Sparse group of models).

**Referee comment #1 (by Loris Foresti)**

**Main comments**

 The forecast verification is well done, but in my opinion it should include a verification of the statistical properties of the advected rainfall fields to understand the degree of numerical diffusion, which can be a major problem in precipitation nowcasting if not properly handled. Such effect usually leads to an undesired smoothing of the precipitation fields, which reduces the more interesting high rainfall intensities and complicates the inter-comparison of models.

**RESPONSE**: We entirely agree that it would be interesting to verify the statistical properties of the advected rainfall fields. It will be done as suggested in comment #11: using a periodogram of rainfall intensities of advected precipitation fields for different lead times.

**ACTION**: A new figure (Figure 5 of the revised manuscript) has been added to address the issue of numerical diffusion (see also ACTION reg. comment #11). We also investigated the issue of numerical diffusion in Section 4 (Results) and Section 5.2 (Advection schemes properties and effectiveness) in the revised manuscript.

- As the paper presents new optical flow and advection techniques, it must include some additional figures showing examples of motion fields and precipitation nowcasts, e.g.:
  - 2.1. A multi-panel figure with vector plots of the motion fields retrieved by the different methods overlaid on top of radar images (for example one "rotational" precipitation event).
  - 2.2. A multi-panel figure showing examples of observed and nowcasted precipitation fields at different lead times, e.g. 30 or 60 minutes. This would be very useful to

understand the quality and realism of the advected rainfall fields, and check whether there are any artefacts due to numerical diffusion and interpolation processes.

**RESPONSE**: We will add requested figures in the revised version of the manuscript.

**ACTION**: A new figure (Figure 5 of the revised manuscript) has been added to show examples of motion fields as well as nowcasts for different models and lead times.

3. Some statements in the literature review are a bit imprecise and could be improved.

**RESPONSE**: We will revise the literature review in the introductory section based on several referee comments (comments #5, #6, #7, and #24 by Dr. Foresti, #1.2 and #2 by Dr. Pulkkinen, #1, #2, and #3 by Dr. Uijlenhoet).

**ACTION**: We have updated the introductory section of the revised manuscript in accordance with the reviewer's comments (see also the actions related to other reviewers' comments: #5, #6, #7, and #24 by Dr. Foresti, #1.2 and #2 by Dr. Pulkkinen, #1, #2, and #3 by Prof. Uijlenhoet).

**Specific comments**

4. Page 1, line 3, Page 2, line 14. "extrapolate the motion" -> "extrapolate the radar echoes". The motion field is usually kept fixed and only the radar echoes are extrapolated, although in some cases it may be beneficial to extrapolate the motion field together with the precipitation echoes.

**RESPONSE**: We suggest to rephrase this to "[...] and then to displace the precipitation field to the imminent future (minutes to hours) based on that motion, [...]". **ACTION**:

1. We rephrased the corresponding sentence in the abstract of the revised version (Page 1, lines 2-5) as follows:

"[...] A common heuristic prediction approach is to track the motion of precipitation features from a sequence of weather radar images, and then to displace the precipitation field to the imminent future (minutes to hours) based on that motion, assuming that the intensity of the features remains constant ("Lagrangian persistence"). [...]"

2. We rephrased the corresponding sentence in the introductory section of the revised version (Page 2, lines 17-18) as follows:

"[...] In the second step, we use that velocity field to advect the most recent rain field, i.e. to displace it to the imminent future based on its observed motion. [...]"

5. Page 2, line 4. The cited approaches (analogue, local Lagrangian and stochastic) were mentioned in the context of probabilistic precipitation nowcasting. They all provide empirical estimates of the probability density function in different ways. Please update accordingly.

**RESPONSE**: Please see response to comment #7.**

ACTION: Please see action to comment #7.

6. Page 2, lines 5-6. Foresti et al. (2015) did not use the correlation coefficient as a measure of similarity to retrieve the analogues (as done e.g. by Atencia et al., 2015), but

rather the Euclidian distance in the space of principal components. Please adjust the statement.

**RESPONSE**: Please see response to comment #7. **ACTION**: Please see action to comment #7.**

7. Page 2, lines 8-10. I think there is some confusion about the definition of "local Lagrangian method". The cited paper (Foresti et al., 2015) follows the definition of Germann and Zawadzki (2004), which defines the "local Lagrangian" as one possible method to derive a probabilistic nowcast. This is achieved by collecting the precipitation values upstream in a local neighbourhood, whose size is increased as a function of lead time.

**RESPONSE**: Comments #5, #6, and #7 are related to one paragraph (Page 2, lines 4-9). We will rewrite the whole paragraph in accordance with the referee's suggestions and try to make the main message of this paragraph (classification of methods used for radar-based precipitation nowcasting) clearer.

We suggest to rephrase the corresponding paragraph to:

"A variety of radar-based precipitation nowcasting techniques can be classified on three major groups based on assumptions we make regarding precipitation field characteristics (Germann and Zawadski, 2002). The first group -- climatological persistence -- provides nowcasts by using climatological values (mean or median). The second group -- Eulerian persistence -- is based on using the latest available observation as a prediction, and is thus independent from the forecast lead time. The third group -- Lagrangian persistence -- allows the extrapolation of the most recent observed precipitation field under the assumption that the motion field is persistent (Germann and Zawadzki, 2002; Woo and Wong, 2017). In addition, we can classify nowcasting methods based on introduced prediction uncertainty: In contrast to deterministic approaches, ensemble nowcast attempt to account for predictive uncertainty by including different realizations of the motion field and the evolution of rainfall intensity itself. In this study, we focus our model development around the group of Lagrangian persistence models which provide deterministic precipitation nowcasts. Yet, the unified availability of different tracking and extrapolation techniques in the rainymotion library could directly be used to construct ensembles that account for the uncertainty of rainfield displacement."

**ACTION**: We have updated the corresponding paragraph in the introductory section (Page 2, lines 4-13) in accordance with the proposed solution in our response:

"[...] A variety of radar-based precipitation nowcasting techniques can be classified into three major groups based on assumptions we make regarding precipitation field characteristics (Germann and Zawadski, 2002). The first group -- climatological persistence -- provides nowcasts by using climatological values (mean or median). The second group -- Eulerian persistence -- is based on using the latest available observation as a prediction, and is thus independent from the forecast lead time. The third group -- Lagrangian persistence -- allows the extrapolation of the most recent observed precipitation field under the assumption that intensity of precipitation features and the motion field are persistent (Germann and Zawadzki, 2002; Woo and Wong, 2017). In addition, we can classify nowcasting methods based on how predictive uncertainty is accounted for: In contrast to deterministic approaches, ensemble nowcasts attempt to account for predictive uncertainty by including different realizations of the motion field and the evolution of rainfall intensity itself (Berenguer et al., 2011). In this study, we focus our

model development around the group of Lagrangian persistence models which provide deterministic precipitation nowcasts. [...]"

8. Page 3, line 1. I fully agree that optical flow libraries have been around for long, but they cannot be directly applied for the retrieval of radar echo motion without important adaptations and tests. For example, they must be tuned to represent the typical range of advection speeds of real precipitation fields, they must be spatially dense and extrapolate well also in regions without precipitation, etc. This is why papers like yours are important contributions to make the necessary adaptations and tests.

**RESPONSE**: We agree that the original manuscript does not sufficiently address how, on the one hand, parameters of different optical flow techniques affect the specific problem of precipitation field tracking, and, on the other hand, how the results of different optical flow techniques might need further post-processing in order to enhance their usefulness for the extrapolation step (e.g. filling or interpolating zero velocities that might occur in regions of zero rainfall). Given that we also introduce further optical flow/tracking as well as extrapolation techniques (comments #9 and #12 by Dr. Foresti, #4-7 by Dr. Pulkkinen) in the revised version of the manuscript and the rainymotion library, the revised manuscript will address these requirements more precisely and comprehensively.

**ACTION**: We have added statements which clarify the possibility to optimize the rainymotion models parameters in order to provide better (in terms of verification efficiency) nowcasts or to represent the typical range of advection speeds of real precipitation fields as follows:

- Page 3, lines 7-12: "[...] That is all the more surprising since open source implementations of fundamental optical flow algorithms (Brox et al., 2004; Bruhn et al., 2005b) have been around for up to 20 years -- with the OpenCV library (https://opencv.org) just being the most widely known. Such libraries provide efficient implementations of various optical flow algorithms for a vast number of research and application contexts. Yet, none can be applied in the QPN context out of the box -without the need to address additional and specific challenges such as underlying assumptions and constraints of velocity fields, pre- and postprocessing steps, or model parameterization and verification. [...]"
- Page 3, lines 32-33; Page 4, line 1: "[...] However, the rainymotion library provides an opportunity to investigate how different optical flow model parameters can affect nowcasting results or how they can be tuned to represent, e.g. the typical range of advection speeds of real precipitation fields. [...]"
- 9. Page 3, line 8. Page 4, line 24. It would be interesting to know why you decided not to include in the list of benchmark extrapolation techniques the backward-in-time semi-Lagrangian scheme, which is generally accepted to be the most appropriate method (Germann and Zawadzki, 2002). The forward scheme is known to produce holes in the precipitation field in presence of divergent vectors, which need to be interpolated. This inevitably leads to additional numerical diffusion.

**RESPONSE**: We originally implemented the forward scheme because it is more intuitive to advect the precipitation field "forward in time" and "downstream in space". Based on the referee's comment, though, we decided to complement the revised version of the rainymotion library with a backward method for the optical flow calculation.

On that basis, we repeated our benchmark experiments by using the backward scheme both for the Dense (constant-vector) and DenseRotation (semi-Lagrangian) models. Results show that the backward scheme performs slightly better for low rainfall intensity rates (under 0.5 mm/h) and longer lead times (from 30 minutes). For rainfall intensity rates over 0.5 mm/h and shorter lead times (up to 30 minutes) there are no significant differences between both schemes. Based on the new results we decided to implement the backward scheme as a default option for precipitation motion field calculation in the revised version of the rainymotion library. We will update the revised version of the manuscript and the supplementary material in accordance with the new results.

However, we also want to note that the intercomparison of different advection schemes provided in Germann and Zawadzki (2002) cannot, in our opinion, be interpreted in a way that "backward-in-time semi-Lagrangian scheme [...] is generally accepted to be the most appropriate method". In the corresponding paper, the forward-in-time scheme is concerted with a gaussian redistribution of advected rainfall in contrast to the interpolation used for backward-in-time scheme. In our library, we adapt the same "interpolate only once" idea of Germann and Zawadzki (2002) -- regardless of the direction used for velocity field (optical flow) calculation that allows intercomparison of forward and backward schemes in a similar setting. Although the new results of our intercomparison are consistent with the referee's statement on the backward scheme being superior (for low rainfall intensities and longer lead times), in our opinion further research is needed to compare the efficiency of different implementations in detail.

**ACTION**: We have updated Section 2 of the revised manuscript and the corresponding sections in the Supplementary information (Sections S4-S6), based on the results of a new benchmarking experiment with both forward and backward schemes as well as with both the constant-vector and the semi-Lagrangian advection schemes, and also modified the rainymotion library accordingly.

10. Page 3, line 26. I cannot understand properly why you mention the concept of scale-dependence in the context of local LK methods. Please explain how local optical flow techniques account for scale-dependence.

**RESPONSE**: Our intention was to highlight that for the Sparse group of rainymotion's tracking models we use distinct "corners" instead of storm cells -- this eliminates the need to specify arbitrary and scale dependent characteristics of "precipitation features" while the identification of "corners" depends only on the gradient sharpness in a cell's neighborhood. Of course this will not solve the issue of scale-dependence of the average motion itself. We will clarify these aspects in the revised manuscript.

**ACTION**: We have updated the corresponding paragraph of Section 2.1 (The Sparse group) as follows (Page 4, Lines 5-8):

"[...] That approach is less arbitrary and scale dependent and thus more universal than classical approaches that track storm cells as contiguous objects (e.g. Wilson et al., 1998) because it eliminates the need to specify arbitrary and scale dependent characteristics of "precipitation features" while the identification of "corners" depends only on the gradient sharpness in a cell's neighborhood. [...]"

11. Page 4, line 5. I am a bit worried that the use of warping and interpolation of discontinuities in the advected radar field can lead to serious numerical diffusion effects.

The most appropriate method to test this issue is to compute the Fourier spectrum of the original and advected fields to check whether there is loss of power at the high spatial frequencies (see Fig. 10 in Germann and Zawadzki, 2002). A simpler approach would be to compare the histogram of nowcasted rainfall fields at different lead times with the one of the last observed radar image. The variance and histogram should be conserved during the extrapolation.

**RESPONSE**: We will update the revised version of the manuscript with figures that describe the level of numerical diffusion for the different models by using the nowcasts' power spectral density for different lead times.

**ACTION**: We have added a new figure (Figure 5 of the revised manuscript) to describe the level of numerical diffusion for different models and lead times (0, +30, and +60 minutes) based on the power spectral density of the nowcasts. We also discussed those results regarding the relevance of numerical diffusion effects in Section 4 (Results) and Section 5.2 (Advection schemes properties and effectiveness).

12. Page 4, line 26. I agree that the constant-vector approach does not explicitly allow to account for rotation. However, if the advection is applied recursively in short time steps the rotation can be approximated by a set of short straight lines (at the cost of stronger diffusion). Despite this fact, I believe that a good implementation of the semi-Lagrangian scheme should consistently give better (or comparable) results than the constant-vector approach.

**RESPONSE**: We agree that an accurate implementation of the semi-Lagrangian scheme should yield a skill that is at least equivalent to the constant-vector approach. We have found two possible reasons why our original implementation did not achieve that: 1. Errors in the estimation of motion fields (e.g. with anomalies, artefacts etc.) could affect the forecast in the semi-Lagrangian advection scheme more than in the constant-vector scheme, since displacement vectors from regions of higher uncertainty might be "activated" more frequently; 2. Higher complexity of semi-Lagrangian scheme implementation which involves interpolation on two levels: when we advect each pixel and try to find the new velocity vector for any new pixel location, and during the final interpolation of intensities.

We attempted to address the estimation of field motion using the (local) Farnebäck optical flow method by implementing a variational refinement procedure to smooth the velocity field, and to get rid of spurious velocities, and by implementing different global optical flow methods that usually provide more smooth and robust motion fields (see also comment #4.1 and #7 by Dr. Pulkkinen). As a result, we included both the variational refinement and different global methods for the tracking step in the rainymotion library, and included these approaches in our benchmarking experiments.

According to these new results, the implementation of the Dense Inverse Search (DIS) global optical flow method (Kroeger et al., 2016) provides better results than the Farnebäck method with variational refinement and other global methods such as DeepFlow (Weinzaepfel et al., 2013) and PCAFlow (Wulff and Black, 2015). Based on these new findings, we decided to use the DIS method as a default method for the precipitation motion field calculation in the revised version of the rainymotion library. We also found that using the DIS method, our results show no significant difference between Dense and DenseRotation models. That confirms the strong influence of motion field estimation on the performance of the DenseRotation model.

We will update manuscript accordingly and show the intercomparison of of different optical flow methods in the supplementary material.

Kroeger, T., Timofte, R., Dai, D., & Van Gool, L. (2016, October). Fast optical flow using dense inverse search. In *European Conference on Computer Vision* (pp. 471-488). Springer, Cham. Weinzaepfel, P., Revaud, J., Harchaoui, Z., & Schmid, C. (2013). DeepFlow: Large displacement optical flow with deep matching. In *Proceedings of the IEEE International Conference on Computer Vision* (pp. 1385-1392).

Wulff, J., & Black, M. J. (2015). Efficient sparse-to-dense optical flow estimation using a learned basis and layers. In *Proceedings of the IEEE Conference on Computer Vision and Pattern Recognition* (pp. 120-130).

**ACTION**: We have updated Section 2 of the revised manuscript and the corresponding section in the Supplementary Information with the new results of benchmarking different global optical flow algorithms and the modified local Farnebäck algorithm. After introducing various changes (in terms of the tracking step, but also with regard to the implementation of the Dense models' extrapolation step, the Dense and the DenseRotation models provide, in effect, the same skill for the selected events and over the lead time of one hour. We also discuss the corresponding differences between implemented advection schemes in Section 5.2 (Advection schemes properties and effectiveness).

13. Page 4, line 29. Also here I would study the effect of numerical diffusion caused by the interpolation. Numerical diffusion can also have undesired consequences when comparing (benchmarking) different nowcasting models. In fact, a precipitation nowcast that loses power at the high spatial frequencies will be generally smoother. This behavior will be rewarded in terms of some verification scores (in particular the MAE/RMSE), which affects the comparison with other models. A fair comparison of different nowcast systems should be done at similar spatial scales, for example using Fourier or wavelet decompositions.

**RESPONSE**: Please see response to comment #11.

ACTION: Please see action to comment #11.

14. Page 6, line 10. "programmatic realization" is a strange expression.

**RESPONSE**: We will rephrase to "In this study we used the RV product data as an operational baseline and did not re-implement the underlying algorithm itself."

**ACTION**: We have revised the corresponding statement in accordance with the statement proposed in the response above (Page 7, lines 20-21).

15. Page 6, line 31. "rainfall depth product". Is it the instantaneous intensity in mm/hr or an accumulation?

**RESPONSE**: The RY product represents rainfall depth in mm for a five minute interval which is however derived from an instantaneous intensity considered representative for that interval.

ACTION: No specific action is needed.

16. Page 6, line 23. It would be very interesting to move the CSI verification at a threshold of 5 mm/hr from the supplementary material to the actual paper. These rainrates are the ones that are relevant to trigger warnings for severe weather.

**RESPONSE**: We will update Figure 6 to represent the CSI for the threshold of 1 mm/h and Figure 7 to represent the CSI for the threshold of 5 mm/h.

**ACTION**: Figure 7 in the revised version of manuscript represents the CSI for a threshold of 1 mm/h, and Figure 8 represents the CSI for a threshold of 5 mm/h.

17. Page 8, lines 5-10. You are correct. Detailed motion fields provide better skill at short lead times, while smoother motion fields are more adapted for longer lead times. Similarly to precipitation fields, the motion fields also have an intrinsic predictability (persistence). This can be exploited by gradually smoothing the motion field in a way that is consistent with its predictability.

**RESPONSE**: We agree. In fact, users can use the library to implement such ideas.**

ACTION: No specific action is needed.

18. Page 8, line 14. All the proposed solutions to the problem of low predictability at convective scales are based on the optical flow and are all valid options. However, precipitation, and in particular the one of convective nature, has a large unpredictable component that we will likely never be able to predict. Therefore, the nowcasting community needs to admit the incapability of providing accurate deterministic precipitation forecasts and find ways to estimate and communicate the inherent uncertainty. I am glad that you presented this issue in the conclusion at page 9, lines 22-25, but it would be a good idea to make this point stronger.

**RESPONSE**: We will try to emphasize this point in the revised version of the manuscript.

**ACTION**: We have updated the corresponding paragraph of the Section 6 (Summary and conclusions) to clarify the point of communicating inherent nowcasting uncertainties (Page 11, lines 26-29):

"[...] Admittedly, deterministic nowcasts in a Lagrangian framework do neither account for precipitation intensity dynamics nor for the uncertainties in representing precipitation field motion. At least for the latter, the rainymotion library provides ample opportunities to experiment with forecast ensembles, based on various tracking and extrapolation techniques [...]"

19. Page 9, line 20-21. I also believe that we should not discard the Sparse models. One possibility is to make them "dense" by interpolating the motion vectors before applying the advection scheme (hopefully semi-Lagrangian).

**RESPONSE**: We thank referee for this comment. More generally, future research should analyse in more detail which steps in our Sparse model chain contribute the most uncertainty. We still think the combination of Sparse optical flow and warping is very efficient and promising, but should be understood better.

ACTION: No specific action is needed.

20. Figures 1 and 2. These are extremely clean and nice presentations of the methods.

**RESPONSE**: We thank referee for this comment.

ACTION: No specific action is needed.

21. Figure 3. You may add in the caption that the figure shows the forward-in-time semi-Lagrangian method.

**RESPONSE**: Figure 3 caption will be updated accordingly.

**ACTION**: We have updated Figure 3 of the revised manuscript to illustrate the four implemented advection methods (forward/backward constant vector, forward/backward semi-Lagrangian).

22. Figure 5. You may consider writing a more descriptive figure caption, e.g. "Verification of the different optical flow based nowcasts in terms of MAE for 11 precipitation events over Germany".

**RESPONSE**: Figure 5 caption will be updated accordingly.

**ACTION**: We have updated the captions of Figures 6-8 in the revised manuscript accordingly.

23. Page 7 line 15, Figures 6-7. Is there an explanation on why the RADVOR nowcasting method performs poorly in the first 5-10 minutes? The effect seems quite systematic and I have a hard time explaining it with the faster movement of precipitation fields.

**RESPONSE**: We briefly described the possible reasons of this RADVOR behavior on Page 7 lines 14-15 and Page 8 lines 3-8. In our opinion, the use of smoothed displacement fields that focus on a large scale motion patterns particularly cause a loss of skill in the RV product for the first 5-10 minutes. We will update the corresponding paragraph of Section 5.1 (Model comparison) to make that point clearer.

**ACTION**: We have updated Section 5.1 according to the new results.

24. Page 9, lines 27-30. With respect to the use of open source libraries to promote the developments in the field of nowcasting, you could also mention how you would imagine the contribution from rainymotion to the developments of other projects, as for example the probabilistic nowcasting library pysteps (https://pysteps.github.io/). In my opinion, any improvement in optical flow methods, e.g. using the rainymotion library, will also have a positive impact on the quality of probabilistic nowcasts. This could represent an interesting synergy between the two libraries, in line with the open source philosophy.

**RESPONSE**: At the moment of paper submission (6th July), no reference to pySTEPS was known to us (first commit on GitHub from 9th July). We will update the introductory section to add a reference to pySTEPS, but we will also include the perspectives mentioned by the referee in the "Summary and conclusions".

**ACTION**: We have updated Section 6 (Summary and conclusions) to highlight PySTEPS and the importance and perspectives of open source software for advancing the field of radar science as follows (Page 12, lines 5-10):

"[...] Recent studies show that open source community-driven software advances the field of weather radar science (Heistermann et al., 2015a, b). Just a few months ago, the pySTEPS (https://pysteps.github.io) initiative was introduced "to develop and maintain an easy to use, modular, free and open source python framework for short-term ensemble prediction systems." As another evidence of the dynamic evolution of QPN research over the recent years, these developments could pave the way for future synergies between the pySTEPS and rainymotion projects -- towards the availability of open, reproducible, and skillful methods in quantitative precipitation nowcasting. [...]"

**Referee comment #2 (by Seppo Pulkkinen)**

**Relation to previous work and literature review**

- 1. There are two important classes of optical flow methods that are only briefly mentioned or not mentioned at all:
  - 1.1. In the variational methods, a smoothness constraint is added to the optical flow equations and they are solved "globally" over the whole domain. The key practical difference to the "local" methods, such as Farnebäck and Lucas-Kanade is that the motion field is automatically filled to areas of no precipitation.
  - 1.2. In the spectral methods, the Fourier transform is applied to the inputs and the optical flow equations are solved in the spectral domain. The authors could add a citation to [3].

[3] E. Ruzanski, V. Chandrasekar and Y. Wang, The CASA Nowcasting System, Journal of Atmospheric and Oceanic Technology, 28(5), 640-655, 2011.

**RESPONSE**: We thank referee for the clarification. We will add the corresponding methods and references to the introduction of the revised version of the manuscript, particularly since we added a variational approach and several global methods to the rainymotion library and our benchmark experiment - please see our response to comment #12 of Dr. Foresti.

**ACTION**: We have updated the corresponding paragraph of the introductory section (Page 2, lines 29-31) to account for the aforementioned group of optical flow methods: *"[...] There is also a distinct group of spectral methods where the Fourier transform is applied to the inputs, and an OFC resolves in the spectral (Fourier) domain (Ruzanski et al., 2011). [...]"*

2. There are several widely used optical flow algorithms developed in the machine vision literature. The authors could cite the Brox and CLG algorithms ([1] and [2]). These have also publicly available C implementations (see the IPOL journal).

 T. Brox, A. Bruhn, N. Papenberg and J. Weickert, High Accuracy Optical Flow Estimation Based on a Theory for Warping, ECCV 2004: 8th European Conference on Computer Vision, Prague, Czech Republic, May 11-14, 2004. Proceedings, Part IV, 25-35, 2004.
 A. Bruhn, J. Weickert and C. Schnörr, Lucas/Kanade Meets Horn/Schunck: Combining Local and Global Optic Flow Methods, International Journal of Computer Vision, 61(3), 211-231, 2005.

**RESPONSE**: We will add those references to the paragraph where we mention only openCV as an open software library with optical flow algorithms implementation (Page 3 line 1 of the discussion paper).

**ACTION**: The references have been added to the revised version of the manuscript as follows (Page 3, lines 7-10):

"[...] That is all the more surprising since open source implementations of fundamental optical flow algorithms (Brox et al., 2004; Bruhn et al., 2005b) have been around for up to 20 years – with the OpenCV library (https://opencv.org) just being the most widely known. Such libraries provide efficient implementations of various optical flow algorithms for a vast number of research and application contexts [...]"

3. The paper cites to a large number of references where more advanced probabilistic nowcasting methods are described. Therefore, in the third paragraph of Section 6 the authors should be more concrete about future plans to include such features into rainymotion, and not just present ideas of potential improvements. Or will rainymotion be restricted only to deterministic extrapolation nowcasting based on Lagrangian persistence?

**RESPONSE**: We would like to thank the referee for this suggestion, yet we are hesitant whether more detailed perspectives on future developments should be elaborated in the paper. Based on its current design, rainymotion's focus is to track the motion of rainfields and to extrapolate future rainfall on that basis. At the same time, the rather low-level implementation easily allows for the flexibility to manipulate the displaced precipitation fields in order to represent -- stochastically or deterministically -- the dynamics of precipitation intensity. Yet, there are no specific plans to implement such features to the rainymotion, but we will, in the revised version of the manuscript, highlight more explicitly the possibility to include such developments.

**ACTION**: We have updated Section 6 (Summary and conclusions) of the revised manuscript with the elaboration of using the rainymotion for providing ensemble nowcasts as follows (Page 11, lines 28-29):

*"[...] At least for the latter, the rainymotion library provides ample opportunities to experiment with forecast ensembles, based on various tracking and extrapolation techniques [...]"*

**Methodology**

- 4. Precisely speaking the Farnebäck optical flow algorithm is not global or dense and should not be called such. This misuse of terminology originates from the OpenCV library.
  - 4.1. The Farnebäck method is dense only in the sense that it produces gridded output instead of motion vectors for sparse feature points as Lucas-Kanade does. If you look at the paper of Farnebäck, the method is formulated as local feature matching, where the solution of the optical flow equations is done by using a polynomial approximation. As a result, the method produces zero motion velocities to areas of no precipitation. You can verify this by plotting motion fields produced by the Farnebäck method.

**RESPONSE**: We thank the referee for pointing that misuse in terminology which we have not been aware of so far. In our paper we use the term "local" and "sparse" in the sense that these methods provide motion vectors at specific locations only. In contrast, we use "global" and "dense" for pointing out that the motion vectors are calculated for an every radar image pixel. We will revise the paper in a way that terminology is both accurate and easy to understand. As a possible solution we propose to change "local" and "global" to "sparse" and "dense" in the revised version of the manuscript and provide a more detailed description of what we consider as "sparse" and "dense" models in the Section 2 (Model). Furthermore, we have actually added global optical flow techniques to the set of tracking models (please see our response to comment #12 of Loris Foresti), so the revised manuscript version will explicitly address the issue of global vs. local optical flow.

**ACTION**: We have revised the manuscript regarding used terminology based on provided suggestions. Since various truly global optical flow techniques have been incorporated into the

rainymotion library and then extensively benchmarked (please see our response to comment #12 of Loris Foresti), we explicitly address the issue of global vs. local optical flow in the revised version of the manuscript.

4.2. It follows from the above that when a pixel is advected into area of no precipitation and a new motion vector is taken at that location (as in the DenseRotation method), it's motion to stops at the boundary. This could explain why Dense has in many cases better performance than DenseRotation.

**RESPONSE**: At the time of submission of this manuscript and thus when the results for this paper had been produced, the implementation of the extrapolation algorithm of the dense optical flow models did in fact not account for the case that pixels are advected into regions of zero velocities. In the meantime, however, we have revised the algorithm so that zero velocities are discarded and replaced by interpolation, and the results will be updated accordingly. Yet, the hypothesis that the insufficient treatment of zero velocities was responsible for the Dense model outperforming the DenseRotation model could not yet be corroborated (please also refer to our response to comment #12 of Dr. Foresti).

**ACTION:** The manuscript and the rainymotion library were substantially revised, particularly regarding the advection schemes for the Dense group of models, based on the corresponding reviewer suggestions. Please also see action to comment #12 of Dr. Foresti.

5. In Germann and Zawadzki (2002), the authors conclude that the backward semi-Lagrangian has better performance than the forward method. In fact, a majority of existing nowcasting methods use the former that is widely regarded as the best approach. However, here the authors use only the latter. If possible, the authors could also implement the backward method and include it in the performance comparison.

**RESPONSE**: Dr. Foresti in his comment #9 also raised this issue. Please refer to our answer there.

ACTION: Please see action to comment #9 of Dr. Foresti.

6. Using the backward method would require filling the gaps in the motion field on areas of no precipitation. Otherwise, no precipitation would be advected into areas where it does not exist at the nowcast start time. A simple distance-weighted interpolation should be sufficient for this purpose. For the above reason, using gap-filling would also improve the performance of the forward semi-Lagrangian method.

**RESPONSE**: In the revised version of the rainymotion library we implemented the referee's suggestion of "[...] filling the gaps in the motion field on areas of no precipitation" by utilizing inverse distance weighted interpolation to fill zero-gaps in the motion field. However, the benefit of this implementation on the performance of the forward semi-Lagrangian method (the DenseRotation model) is not so distinct probably because of the reasons we highlighted in the response on the comment #12 from Dr. Foresti (motion field estimation errors by the Farnebäck algorithm and additional interpolation).

**ACTION**: The manuscript and the rainymotion library have been revised in accordance with the above response.

7. Note that the gap-filling is automatically done in the variational methods without the need for separate post-processing of the motion field. Therefore, such methods are truly dense and global. The authors could consider implementing a variational method and include it in the performance comparison.

**RESPONSE**: We thank the referee for his recommendations regarding the implementation of variational optical flow models in the rainymotion library. We incorporated global optical flow methods which are available in opencv library as additional options for motion field calculation in the rainymotion library (see also our response to the comment #12 from Dr. Foresti), verified their skill for nowcasting and have to conclude that using more advanced global optical flow methods advances an efficiency of a semi-Lagrangian advection scheme. Based on the new obtained results we decided to replace the Farnebäck method by the global Dense Inverse Search (DIS, Kroeger et al., 2016) as a default tracking option. We will also update the supplementary material with intercomparison results of different optical flow methods. Kroeger, T., Timofte, R., Dai, D., & Van Gool, L. (2016, October). Fast optical flow using dense inverse search. In *European Conference on Computer Vision* (pp. 471-488). Springer, Cham.

**ACTION**: The manuscript has been revised according to the above response.

**Software library**

8. Sections 2.4 and 3: Is the library restricted only to using the DWD data? Please add discussion about how to use the library with other file formats? For instance, by using wradlib this should be easily done because it supports a large number of different formats.

**RESPONSE**: There is no restriction in using different data formats because of rainymotion works directly with numpy arrays, and the data preprocessing routine is fully on the user-side. There is a set of available open software libraries for radar data reading and preprocessing (the list available on https://openradarscience.org/). We will add the corresponding information to the Section 2.4 (The rainymotion Python library).

**ACTION**: We have updated Section 2.4 (The rainymotion Python library) in accordance with provided response as follows (Page 6, lines 28-31):

"[...] Since the rainymotion uses standard format of numpy arrays for data manipulation, there is no restriction in using different data formats which can be read, transformed, and converted to numpy arrays using any tool from the set of available open software libraries for radar data manipulation (the list is available on https://openradarscience.org). [...]"

**Verification**

9. Section 2.6: MAE could be computed conditionally over those pixels where both the nowcast and the verifying observation exceed the detection threshold. Otherwise, there would be overlap with the CSI statistic as both penalize incorrect forecasts of precipitation/no precipitation.

**RESPONSE**: In our study we decided to use MAE as a score from a continuous category and implement it directly without making specific thresholds (like we do for categorical category of

verification scores). In our opinion, this admittedly arbitrary decision of using different verification score categories helps to represent a diversity of obtained results.

ACTION: No specific action is needed.

- 10. A large number of CSI and MAE statistics are shown for different lead times. There could be more analysis of the results.
  - 10.1. There is no indication about what can be considered as a good CSI or MAE value for the nowcast to be usable. Can you give some thresholds?

**RESPONSE**: At the best of our knowledge, there is no convention regarding what to consider as "good" or "bad" for any verification metric commonly used in radar-based QPN. For our benchmarking experiment, the focus is on the differences of scores between the different models, not on their absolute values.

**ACTION**: No specific action is needed.

10.2. The differences between the methods (excluding Persistence) are relatively small in terms of CSI and MAE statistics. Based on such differences, the authors should be more careful when claiming that some method is better than another. For instance the maximum mean difference between Dense and DenseRotation is only 0.01 according to Table 3.

**RESPONSE**: Table 3 represents statistics which are averaged over all the analyzed events and two lead time periods (5--30, and 35--60) and primarily highlight the difference between the Dense group of rainymotion models (Dense and DenseRotation) and and the RV product (as mentioned on the Page 7, lines 27--31) -- which is more distinct than the difference between Dense and DenseRotation models themselves. For the verification procedure we also carried out the Student's independent two-sample *t*-test to find whether differences between mean CSI and MAE values for the specific lead times are significant or not (not shown in the manuscript). We found that the results of the visual inspection of the verification plots are well consistent with the formal statistical evaluation: if there is a clear difference in the plots, it is typically significant in a statistical sense.

We will update Table 3 with the new results and adjust our statements about considering one model better/worse than another correspondingly.

**ACTION**: Table 3 of the revised manuscript has been updated with the new verification results.

- 11. Figures 5-7 and p. 9, lines 19-21. The authors should indeed take a closer look on why the performance of the sparse methods is poor. Some comments about this:
  - 11.1. The relevant parameter here is the number of features used in the tracking and nowcasting. If this number is too small, the motion vectors of the features are not representative of the large-scale motion field. Can you check this by adjusting the thresholds in the feature detector?
  - 11.2. In addition, can you specify somewhere how many feature points are used with the sparse methods because this is a key parameter?
  - 11.3. Another point missed in the paper is that the corner detector tends to pick features that have high intensities and gradients. Therefore, a very careful quality control is needed to ensure that the features are precipitation and not some

random artefacts in the radar data. Can you be sure that the quality control is sufficient?

11.4. Even if the features are precipitation, they represent small-scale phenomena that can have very different motion from the large-scale advection field. Thus, the representativity of such features can be very poor.

**RESPONSE**: We agree with the referee that the sensitivity of the Sparse group of models to specific key parameters needs to be investigated more closely. Yet, we consider such an analysis beyond the scope of this study. Another study is underway that specifically and systematically focuses on the error of the forecast location of detected features based on a vast set of tracking and extrapolation techniques, and including different parameterisations as mentioned by the referee (such as the maximum number of features detected, or different approaches to filter spurious or non-representative velocities at small spatiotemporal scales). In the present manuscript under discussion, however, our aim is to present two basic and open architectures of nowcasting models based on optical flow which can serve as a baseline for future developments - as part of the rainymotion library itself or in combination with the library, and to demonstrate that these are skillful. Still, the parameters of the Shi-Tomasi corner detector provide us a possibility to control the maximum number of features, their quality (which is based on the minimal eigenvalue) and a minimum euclidean distance between the nearest identified points. A calibration of these parameters had been performed on different events and the most robust values had been set up as default parameters as follows: maximum number of features --200; quality level -- 0.2 (the corners with the quality measure less than the product of quality level and minimal eigenvalue will be rejected); minimum euclidean distance -- 7 pixels (the corners which have stronger neighbors in a neighborhood less than 7 pixels will be rejected). As for quality control of the actual radar data, we rely on the DWD's processing workflow that produces the RY product and which eliminates vast parts of spurious echoes. Yet, even in the presence of residual static or dynamic clutter, the tracking algorithm has proven to be robust against producing zero velocities.

**ACTION**: We have updated Section 5.1 (Model comparison) with the comparison of the Dense and Sparse group of the rainymotion in accordance with the reviewer suggestions as follows (Page 10, lines 5-11):

[...] Despite their skill over Eulerian persistence, the Sparse group models are significantly outperformed by the Dense group models for all the analyzed events and lead times. The reason for this behaviour remains yet unclear. It could, in general, be a combination of errors introduced in corner-tracking and extrapolation as well as image warping as a surrogate for formal advection. While the systematic identification of error sources will be subject to future studies, we suspect that the the local features ("corners") identified by the Shi-Tomasi corner detector might not be representative for the overall motion of the precipitation field: the detection focuses on features with high intensities and gradients, the motion of which might not represent the dominant meso- $\gamma$  scale motion patterns. [...]

12. Forecasting the occurrence of precipitation/no precipitation for high intensities is highly relevant for practical applications. Therefore, I would suggest moving the results with the 5 mm/h threshold from the supplementary material to the main paper.

**RESPONSE**: We support referee's recommendation (see also comment #16 by Loris Foresti) and will transfer the corresponding figure from the supplementary to the main paper.

**ACTION**: The new Figure 8 represents the CSI for the threshold of 5 mm/h in the revised version of the manuscript.

**Figures**

13. Since the motion field determination plays a key role in the paper, the authors should show at least one figure with an observed precipitation field and the computed motion field plotted on the same figure. Even better would be a figure showing motion vectors of features and motion fields computed by using different methods.

**RESPONSE**: We agree with the referees' recommendation (see also comment #2 by Loris Foresti) and will add the requested figures to the revised version of the manuscript.

**ACTION**: The new figure (Figure 5) has been added to the revised version of the manuscript to show examples of nowcasts, velocity vectors of features and velocity fields for different models and lead times.

14. Figure 4: Are names of individual functions relevant here? Consider removing them.

**RESPONSE**: In our opinion, it is informative to show the key functions that we used from various libraries in order to put together the main functionality of rainymotion. It illustrates that the combination is, from a technical perspective, not too complex.

**ACTION**: We have updated Figure 4 in the revised manuscript based on the changes in the rainymotion library.

**Minor details**

15. Page 4, lines 3-6 and Figure 1. How exactly is the affine transformation matrix calculated. In particular, is a single matrix estimated for all features or is this done separately for each feature?

**RESPONSE**: The transformation matrix is calculated on the basis of all identified features. We will add this clarification to Section 2.1 (Local optical flow models).

**ACTION**: We have updated the corresponding statement in the revised manuscript (Page 4, lines 22-23).

16. Page 5, line 24. Why the HDF5 file format was chosen? Please add some justification for this.

**RESPONSE**: For all internal projects we use HDF5 database and corresponding file format as an efficient data storage with powerful set of archiving options (i.e. compression rate, chunk size) instead of using default binary files provided by the DWD. However, we propose to remove the reference to HDF5 file format and h5py library because of it is neither integral part of our analysis, nor the rainymotion library, but just a subjective choice we made regarding our research workflow. We will update the Section 3 correspondingly.

**ACTION**: The references to HDF5 file format and h5py library have been removed from the revised version of the manuscript.

17. Page 9, lines 7-9. I don't understand what this means. Can you clarify?

**RESPONSE**: The statement "It might also be considered to combine the warping procedure for the extrapolation step with the Dense optical flow procedure for the tracking step in order to dramatically enhance computational performance" describes the idea to detect corners, then predict the future locations of these corners using the motion field from dense optical flow, and then construct the Affine Transformation Matrix for the warping based on the corner locations at forecast time and lead time  $t_n$ . That way, we would combine the robustness of the dense optical flow technique with the computational efficiency of the warping technique. We will clarify that idea in the revised manuscript.

**ACTION**: The statement under consideration has been updated as follows (Page 11, lines 20-22):

"[...] It might also be considered to combine the warping procedure for the extrapolation step with the Dense optical flow procedure for the tracking step (i.e. to advect "corners" based on a "Dense" velocity field obtained by implementing one of the dense optical flow techniques). [...]"

18. Page 9, line 24. Stochastic accounting
  - 5.2. P.4, I.8: "24 recent radar images" -> "24 most recent radar images".
  - 5.3. P.5, I.20: "models' description" -> "model description".
  - 5.4. P.6, I.25–26: "rainfall rates prediction" -> "rainfall rate prediction".
  - 5.5. P.8, I.6: Insert comma before "which".

**RESPONSE**: Will be fixed.

ACTION: The corresponding issues have been fixed.

6. Is "RV" the same ad "RadVor"?

**RESPONSE**: RADVOR is the entire nowcasting workflow used by the DWD. RV is a main product along that processing chain which is the forecast precipitation depth in five minute intervals over a lead time of two hours. The official main product of RADVOR, though, is the RQ product which is the precipitation depth accumulated over an interval of one hour for a lead time of two hours. It is basically obtained from the RV product, but includes an additional adjustment of the distribution function. In summary, the RV product is the part of DWD's nowcasting chain that is best comparable to our nowcasting products and the best "end product" that is available at an interval of five minutes.

ACTION: No specific action is needed.

7. General: (much) more detailed captions; figures + captions should be as self-contained as possible.

**RESPONSE**: We will update the figure captions to make them more self-contained.

**ACTION**: We have updated captions for figures 6-8 according to comment #22 of Dr. Loresti.

8. Journal (Nature), issue, page numbers missing from reference to Bauer et al. (2015).

**RESPONSE**: We will update the corresponding reference to the Bauer et al. (2015) paper as following:

Bauer, P., Thorpe, A., Brunet G.: The quiet revolution of numerical weather prediction, Nature, 525, 47–55, https://doi.org/10.1038/nature14956, https://www.nature.com/articles/nature14956, 2015.

**ACTION**: The reference to Bauer et al. (2015) has been updated.

**Other changes made in the manuscript**

- 1. All the figures and tables (except Table 1) in the manuscript and the Supplementary Information have been updated. Additionally, new sections of the Supplementary information (Sections S5, S6) have been added based on the new results which have been obtained using the revised version of the rainymotion library.
- 2. Subsections of Section 2 (Models) have been updated to fit the proposed substitution of terminology from local/global optical flow to sparse/dense.
- 3. The rainymotion library source code and documentation have been substantially updated (https://github.com/hydrogo/rainymotion/commits/v0.1).

[revised manuscript text omitted]
://doi.00111016/j.jhydrol.2011.04.033, https://doi.00111016/j.jhydrol.2011.04.033, https://doi.00111016/j.jhydrol.2011.04.033, https://doi.00111016/j.jhydrol.2011.04.033, https://doi.00111016/j.jhydrol.2011.04.033, https://doi.00111016/j.jhydrol.2011.04.033, https://doi.00111016/j.jhydrol.2011.04.033, https://doi.00111016/j.jhydrol.2011016/j.jhydrol.2011016/j.jhydrol.2011016/j.jhydrol.2011016/j.jhydrol.2011016/j.jhydrol.2011016/j.jhydrol.2011016/j.jhydrol.2011016/j.jhydrol.2011016/j.jhydrol.2011016/j.jhydrol.2011016/j.jhydrol.2011016/j.jhydrol.2011016/j.jhydrol.2011016/j.jhydrol.2011016/j.jhydrol.2011016/j.jhydrol.2011016/j.jhydrol.2011016/j.jhydrol.2011016/j.jhydrol.2011016/j.jhydrol.2011016/j.jhydrol.2011016/j.jhydrol.2011016/j.jhydrol.2011016/j.jhydrol.2011016/j.jhydrol.2011016/j.jhydrol.2011016/j.jhydrol.2011016/j.jhydrol.2011016/j.jhydrol.2011016/j.jhydrol.2011016/j.jhydrol.2011016/j.jhydrol.2011016/j.jhydrol.2011016/j.jhydrol.2011016/j.jhydrol.201106/j.jhydrol.201106/j.jhydrol.201106/j.jhydrol.201106/j.jhydr

[revised manuscript text omitted]

Propagate features linearly for every lead time n

t-1 t t+1 t+2 t+3

---

## Referee Report (RR1)

| Page 2, line 1                     | "1-3 hours:" $\rightarrow$ "1-3 hours, for example:"                                                                                                                                                                                                                                                                                                |
|------------------------------------|-----------------------------------------------------------------------------------------------------------------------------------------------------------------------------------------------------------------------------------------------------------------------------------------------------------------------------------------------------|
| Page 2, line 30                    | "the OFC is resolved"?                                                                                                                                                                                                                                                                                                                              |
| Page 2, line 33-35                 | What did Liu et al. (2015) find from their study? Is it better to use LK or HS on satellite images?                                                                                                                                                                                                                                                 |
| Table 1                            | In the caption you should specify the grid size and that the computational time refers to both optical flow and extrapolation for 12 frames (as done in Sect. 5.3). It could be worth separating the computational time of the optical flow from the one of the extrapolation. That would be interesting.                                           |
| Page 6, line 25                    | "predictor" $\rightarrow$ "model" or "assumption"?                                                                                                                                                                                                                                                                                                  |
| Eq. 1                              | Is the MAE integrated over all pixels or conditioned to where either the nowcast or the observations have rain? Please specify.                                                                                                                                                                                                                     |
| Figure 8 and page
8, line 26-27 | Please show the power spectral densities down to double of the grid resolution, i.e. 2 km. Germann and Zawadzki (2002) made the analyses on a 4 km resolution radar grid. That's why they stopped the spectra at 8 km. Now that the "interpolate once" method is implemented in rainymotion, there should be basically no numerical diffusion left. |

---

## Author Response (AR3)

**Response letter**

Dear Referees, dear Editor,

We would like to thank you again for your positive comments and constructive suggestions for the improvement of our manuscript. In this document, we would like to provide our responses to the comments of Loris Foresti and Seppo Pulkkinen including the decision on specific changes in the manuscript. For that purpose, we use the following color code:

black: original referee comment blue: our response

Sincerely, Georgy (on behalf of the authors)

**Referee comments by Loris Foresti**

1. Page 2, line 1: "1-3 hours:" "1-3 hours, for example:" ACTION: Fixed.

2. Page 2, line 30: "the OFC is resolved"? ACTION: Fixed.

3. Page 2, line 33-35: What did Liu et al. (2015) find from their study? Is it better to use LK or HS on satellite images?

ACTION: The corresponding sentence "Liu et al. (2015) proposed using a local Lucas–Kanade optical flow method (Lucas and Kanade, 1981) independently for each pixel of satellite imagery and compared its performance with a global Horn–Schunck (Horn and Schunck, 1981) optical flow algorithm." in the Introductory section has been updated as follows:

"Liu et al. (2015) proposed using a local Lucas–Kanade optical flow method (Lucas and Kanade, 1981) independently for each pixel of satellite imagery because they found it outperformed a global Horn–Schunck (Horn and Schunck, 1981) optical flow algorithm in the context of precipitation nowcasting from infrared satellite images."

4. Table 1: In the caption you should specify the grid size and that the computational time refers to both optical flow and extrapolation for 12 frames (as done in Sect. 5.3). It could be worth separating the computational time of the optical flow from the one of the extrapolation. That would be interesting.

ACTION: Table 1 has been updated as follows:

Table 1. Overview of the developed nowcasting models and their computational performance. Nowcasting experiments were carried out for one hour lead time in 5 min temporal resolution (12 resulting nowcast frames in total) using the RY radar data (spatial resolution of 1 km, grid size 900x900) and a standard office PC with an Intel® Core TM i7-2600 CPU (8 cores, 3.4 GHz).

| Model name    |
Computational time
(tracking/extrapolation/total), s |
|---------------|-------------------------------------------------------------|
| Sparse        |
0.2 / 1.1 / 1.3                                         |
| SparseSD      |
0.1 / 1.1 / 1.2                                         |
| Dense         |
0.2 / 5.5 / 5.7                                         |
| DenseRotation |
3.2 / 5.7 / 8.9                                         |

The corresponding sentence "The average time for generating one nowcast for one hour lead time (at 5 minute resolution) for the Sparse group is 2-3 s, and for the Dense group is 7-12 s." has been updated as follows:

"The average time for generating one nowcast for one hour lead time (at 5 minute resolution) for the Sparse group is 1.5-3 s, and for the Dense group is 6-12 s."

5. Page 6, line 25: "predictor" -> "model" or "assumption"?

ACTION: In the corresponding sentence the word "predictor" has been substituted by "model".

6. Eq. 1: Is the MAE integrated over all pixels or conditioned to where either the nowcast or the observations have rain? Please specify.

ACTION: The corresponding sentence "... where nowi and obsi are nowcast and observed rainfall rate in the i-th pixel of the corresponding radar image, and n the number of pixels." has been supplemented by the following sentence:

"... To compute the MAE, no pixels were excluded based on thresholds of nowcast or observed rainfall rate."

7. Figure 5 and page 8, line 26-27: Please show the power spectral densities down to double of the grid resolution, i.e. 2 km. Germann and Zawadzki (2002) made the analyses on a 4 km resolution radar grid. That's why they stopped the spectra at 8 km. Now that the "interpolate once" method is implemented in rainymotion, there should be basically no numerical diffusion left.

ACTION: Figure 5 has been updated.

The corresponding part in the text with the analysis of numerical diffusion properties "As had been shown in (Germann and Zawadzki, 2002), the most significant loss of power spectra (lower PSD values) refers to small-scale precipitation patterns in the range of 8 to 64 km, so we constrained the PSD plots to highlight that range. The power spectra show that, compared to the observations, the loss of power is small for all lead times, scales, and models (Sparse and Dense). At least for this example, it appears that both the warping and the "interpolate only once" approaches are successful in limiting the effects of numerical diffusion and thus the loss of power at small scales – at least for lead times up to one hour." has been updated as follows:

"Germann and Zawadzki (2002) showed that the most significant loss of power (lower PSD values) occurs at scales between 8 to 64 km. They did not analyse scales below 8 (23) km because their original grid resolution was 4 km. We extended the spectral analysis to consider scales as small as 21 km. Other than Germann and Zawadzki (2002), we could not observe any

substantial loss of power between 8 and 64 km, yet Figure 5 shows that both Dense and Sparse models consistently start to lose power at scales below 4 km. That loss does not depend much on the nowcast lead time, yet, the Sparse group of models loses more power at a lead time of 5 minutes as compared to the Dense group. Still, these results rather confirm Germann and Zawadzki (2002): they show, as would be expected, that any loss of spectral power is most pronounced at the smallest scales, and disappears at scales about 2-3 orders above the native grid resolution. For the investigated combination of data and models, that implies that our nowcasts will not be able to adequately represent rainfall features smaller than 4 km at lead times of up to 1 hour."

**Referee comments by Seppo Pulkkinen**

General comments:

The authors have done a very good and detailed work to address all my previous questions. I still have one comment about how the text is organized.

- The title of Section 2 is "Models" that does not match the content. This is because Section 2 also contains description of the library (Section 2.4) and the verification metrics (Section 2.6). In my opinion, there should be one section devoted to the models, and the above items should be separated from this section. In addition, Sections 3 and 4 are very short. Suggestions for improvement:
- Section 2.4 could be separated into its own main section (e.g. Section 3).

• Sections 2.6 and 3 could be moved into Section 4 that would describe the experiment setup, the used verification metrics, the data and also the results. Also change the title accordingly.

• If Section 2.4 is kept as is, Section 2 could be renamed, for instance, as "Description of the models and the library".

ACTION: We reorganized the manuscript structure by reviewer suggestions as follows:

2. Description of the models and the library

- 2.1 The Sparse group
- 2.2 The Dense group
- 2.3 Persistence
- 2.4 The *rainymotion* Python library
- 2.5 Operational baseline (RADVOR)
- 3. Verification experiments
- 3.1 Radar data and verification events (previously 3)
- 3.2 Verification metrics (previously 2.6)

Minor details:

1. p.1, lines 22-23: Can you add a reference which states that radar-based nowcasting outperforms NWP at short lead times?

ACTION: References have been added:

Berenguer, M., Surcel, M., Zawadzki, I., Xue, M., Kong, F., Berenguer, M., ... Kong, F. (2012). The Diurnal Cycle of Precipitation from Continental Radar Mosaics and Numerical Weather Prediction Models. Part II: Intercomparison among Numerical Models and with Nowcasting. Monthly Weather Review, 140(8), 2689–2705. https://doi.org/10.1175/MWR-D-11-00181.1 Jensen, D. G., Petersen, C., & Rasmussen, M. R. (2015). Assimilation of radar-based nowcast into a HIRLAM NWP model. Meteorological Applications, 22(3), 485–494. https://doi.org/10.1002/met.1479

Lin, C., Vasić, S., Kilambi, A., Turner, B., & Zawadzki, I. (2005). Precipitation forecast skill of numerical weather prediction models and radar nowcasts. Geophysical Research Letters, 32(14). https://doi.org/10.1029/2005GL023451

p. 4, l. 15: More detailed explanation could be added here. Eigenvalues of what? Are the eigenvalues computed by using single pixels or a neighborhood around each pixel?
 ACTION: The corresponding sentence "This detector determines the most prominent corners in the image based on the calculation of the corner quality measure (min(λ 1 , λ 2 ), where λ 1 and λ 2 are corresponding eigenvalues) at each image pixel (see Section S1 of the Supplementary Information for a detailed description of algorithm parameters);" has been updated as follows:

"This detector determines the most prominent corners in the image based on the calculation of the corner quality measure (min( $\lambda 1$ ,  $\lambda 2$ ), where  $\lambda 1$  and  $\lambda 2$  are corresponding Eigenvalues of the covariance matrix of derivatives over the neighborhood of 3 by 3 pixels) at each image pixel (see Section S1 of the Supplementary Information for a detailed description of algorithm parameters);"

Supplementary material:

S5: It is mentioned that "Results show that the DIS optical flow technique outperforms the remaining optical flow techniques while also providing the fastest computation.". However, I could not find any graph or table showing the computation times of different optical flow methods. Can you add this information?

ACTION: The corresponding sentence "Results show that the DIS optical flow technique outperforms the remaining optical flow techniques while also providing the fastest computation." has been updated with the reference to Table S2.

| Optical flow technique | Computational time |
|------------------------|--------------------|
| DIS                    | 19.8 ms ± 1.76 ms  |
| PCAFlow                | 287 ms ± 8.67 ms   |
| Farnebäck              | 715 ms ± 11.9 ms   |
| DeepFlow               | 3.77 s ± 117 ms    |

Table S2. Computational time of different optical flow techniques (mean ± standard deviation of 100 runs)

[revised manuscript text omitted]

Propagate features linearly for every lead time n

t-1 t t+1 t+2 t+3

Figure 1. Scheme of the SparseSD model